# A Sketch-and-Project Analysis of Subsampled Natural Gradient Algorithms

**Gil Goldshlager** [1]   **Jiang Hu** [2]   **Lin Lin** [1][3]

## Abstract

Subsampled natural gradient descent (SNG) has been used to enable high-precision scientific machine learning, but standard analyses based on stochastic preconditioning fail to provide insight into realistic small-sample settings. We overcome this limitation by instead analyzing SNG as a sketch-and-project method. Motivated by this lens, we discard the usual theoretical proxy which decouples gradients and preconditioners using two independent mini-batches, and we replace it with a new proxy based on squared volume sampling. Under this new proxy we show that the expectation of the SNG direction becomes equal to a preconditioned gradient descent step even in the presence of coupling, leading to (i) global convergence guarantees when using a single mini-batch of any size, and (ii) an explicit characterization of the convergence rate in terms of quantities related to the sketch-and-project structure. These findings in turn yield new insights into small-sample settings, for example by suggesting that the advantage of SNG over SGD is that it can more effectively exploit spectral decay in the model Jacobian. We also extend these ideas to explain a popular structured momentum scheme for SNG, known as SPRING, by showing that it arises naturally from accelerated sketch-and-project methods.

## 1. Introduction

Neural networks are increasingly gaining traction as an alternative to traditional numerical solvers, with applications including neural network wavefunctions (NNWs, Carleo & Troyer, 2017) and physics-informed neural networks (PINNs, Raissi et al., 2019). For such scientific machine learning problems, precision is prioritized over generaliza-

tion performance. As a result, natural gradient algorithms can dramatically outperform more standard machine learning optimizers such as gradient descent and Adam (Pfau et al., 2020; Schätzle et al., 2023; Lin et al., 2023; Müller & Zeinhofer, 2023; Dangel et al., 2024). Furthermore, high iteration costs can be mitigated by *subsampling* (Ren & Goldfarb, 2019), leading to efficient subsampled natural gradient (SNG) algorithms that scale to millions of parameters without approximation. This approach was introduced to the NNW community by (Chen & Heyl, 2024) and since this seminal work, SNG has become a workhorse algorithm for NNWs; see (Rende et al., 2024; Lange et al., 2025; Vecsei & Lado, 2025; Sinibaldi et al., 2025; Misery et al., 2026) and many more. SNG has also demonstrated promising results for PINNs (Schwencke & Furtlehner, 2024; Guzmán-Cordero et al., 2025; Jnini & Vella, 2025), though it has not yet been widely adopted.

Despite these developments, subsampled natural gradient algorithms remain poorly understood. Analytically, the key challenge lies in controlling the expectation of an optimization step involving a stochastic gradient that is coupled to the inverse of a stochastic preconditioner. The standard way to address this problem is to introduce a theoretical proxy which breaks the coupling using two independent mini-batches; see Appendix A for details. This approach can lead to detailed convergence rates when the sample size is large enough to accurately estimate the preconditioner, but for smaller sample sizes it produces only generic global convergence guarantees (Ren & Goldfarb, 2019; Roosta-Khorasani & Mahoney, 2019; Bollapragada et al., 2019; Martens, 2020; Yang et al., 2020; Wu et al., 2024). As a result, such analyses fail to provide insight into realistic settings in which the sample size is orders of magnitude smaller than the number of parameters. Moreover, this failure is predictable: for such sample sizes the stochastic preconditioner captures only a tiny subset of the parameter space, making it unlikely to be informative for an independent stochastic gradient. As a result, even for very simple problems the two mini-batch proxy can exhibit drastically different behavior than the realistic, single mini-batch algorithm; see Figure 2 and Appendix I.

Several recent works have hinted at a different perspective (Chen & Heyl, 2024; Schwencke & Furtlehner, 2024; Goldshlager et al., 2024; 2025). These works have viewed the

---

[1]Department of Mathematics, University of California, Berkeley [2]Yau Mathematical Sciences Center, Tsinghua University [3]Lawrence Berkeley National Laboratory. Correspondence to: Gil Goldshlager <ggoldsh@berkeley.edu>.

*Proceedings of the 43rd International Conference on Machine Learning*, Seoul, South Korea. PMLR 306, 2026. Copyright 2026 by the author(s).

| Conceptual lens | Global guarantees? | Small-Sample Insights? | Principled acceleration? | Technical assumption |
|---|---|---|---|---|
| Stochastic preconditioning | ✓ | ✗ | ✗ | Two independent mini-batches |
| Sketch-and-project | ✓ | ✓ | ✓ | Squared volume sampling |

*Figure 1.* Summary of technical contributions relative to previous analyses based on stochastic preconditioning.

SNG update as the minimal-norm solution to a subsampled linear system rather than an estimator for the deterministic natural gradient direction. Along these lines connections have been drawn to randomized block Kaczmarz methods, which are a special case of the sketch-and-project framework of (Gower & Richtárik, 2015a). This direction is promising since sketch-and-project methods use a single sketching matrix for each iteration and admit detailed convergence rates for arbitrary sketch sizes. Nonetheless, it has remained unclear whether this approach can lead to new theoretical insights into SNG, particularly since embracing the sketch-and-project perspective requires avoiding the decoupling tactic that has facilitated previous analyses.

## 2. Contributions

We develop the sketch-and-project approach into a systematic framework for analyzing subsampled natural gradient algorithms. Our starting point is the general parametric optimization problem

$$\min_{\theta} \ \mathcal{L}(v_\theta) \tag{1}$$

where $\theta \in \mathbb{R}^n$ are the parameters, $v_\theta : \Omega \to \mathbb{R}$ is an element of a function space $\mathcal{H}$, and $\mathcal{L} : \mathcal{H} \to \mathbb{R}$ is a loss defined on the function space. For scientific machine learning the underlying science problem is here represented by $\min_v \mathcal{L}(v)$, while the neural network ansatz is represented by $v_\theta$.

In this setting, the availability of high-precision solutions depends on the consistency of the problem, with exact consistency ($v_{\theta*} = \arg\min_v \mathcal{L}(v)$) enabling exact solutions and approximate consistency ($v_{\theta*} \approx \arg\min_v \mathcal{L}(v)$) enabling approximate solutions. To understand the effectiveness of SNG for high-precision scientific machine learning, it is thus natural to study its local convergence properties for consistent problems. To do so cleanly, we note that near the minimizer, any consistent parametric optimization problem is approximated to second order as $\mathcal{L}(v_\theta) \approx \tilde{\mathcal{L}}(\tilde{v}_\theta)$, where $\tilde{\mathcal{L}}$ is a quadratic approximation to $\mathcal{L}$ about $\theta^*$ and $\tilde{v}$ is a linearization of $v$ about $\theta^*$. Motivated by this fact we focus our convergence rate analysis on the resulting model problem:

**Problem 2.1** (Linear least-quadratics). *Let $\mathcal{H} = \mathbb{R}^m$ and*

$$v_\theta = J\theta, \quad \mathcal{L}(v) = \frac{1}{2}v^\top H v - v^\top b \tag{2}$$

*with $H \succ 0$. Then minimize $\mathcal{L}(v_\theta)$ as in (1).*

We call this problem *linear least-quadratics* (LLQ) since it uses a linear ansatz to minimize a quadratic loss function, and we note that the standard linear least-squares is recovered by setting $H = I$. We also note that here and in the rest of this work we assume $\mathcal{H} = \mathbb{R}^m$ (equivalently, $\Omega = \{1, \ldots, m\}$) to simplify the discussion, though we keep our notation general to emphasize the broader applicability of the algorithms. With this context in mind we summarize our contributions as follows:

1. **We provide an approach to analyzing SNG without decoupling the gradient and the preconditioner.** In particular, we show that under a sampling strategy known as *squared volume sampling* (SVS), the expected SNG direction becomes equal to a preconditioned gradient descent step, even in the presence of coupling (Lemma 4.1). This leads to global convergence guarantees when using a single mini-batch of any size to inform each step (Theorem 4.2), and it establishes SVS-SNG as a new theoretical proxy for SNG. The relevance of this construction is supported by numerical experiments which show that SVS-SNG can be a more faithful proxy for realistic SNG algorithms than previous proxies based on independent mini-batches (Figure 2 and Appendix I).

2. **We show that the convergence rate of SNG can be controlled by its sketch-and-project structure.** In particular, under SVS we prove an explicit convergence rate for SNG in the linear least-quadratics setting (Theorem 5.1). The results show that as the sample size is varied, the optimal convergence rate scales as $\alpha/\gamma$, where $\alpha$ is the standard sketch-and-project convergence rate and $\gamma$ is a new quantity related to the second moment of the sketch-and-project step. The appearance of $\alpha$ suggests that the advantage of SNG over SGD is

that it can more effectively exploit spectral decay in the model Jacobian, which is confirmed empirically in Figure 4. Regarding $\gamma$, we provide a preliminary characterization (Proposition 5.3) and numerical evidence which suggests benign behavior (Figure 3), and we defer a thorough analysis to future works. In Appendix G we also provide an extension of Theorem 5.1 to the more general setting when $\mathcal{L}$ is smooth and strongly convex and the problem is possibly inconsistent.

3. **We extend these ideas to explain a popular structured momentum scheme for SNG.** In particular, we show that the subsampled projected-increment natural gradient (SPRING) algorithm of (Goldshlager et al., 2024) arises naturally from the application of accelerated sketch-and-project methods (Theorem 6.1). This suggests that the advantage of SPRING over SNG should be greatest when the underlying sketch-and-project steps converge slowly, such as when the sample size is small; this phenomenon is confirmed empirically in Figure 5.

Overall, our results suggest that when using subsampled natural gradient algorithms to solve scientific machine learning problems, more attention should be paid to the sketch-and-project properties such as $\alpha$ and $\gamma$, and less to the statistical estimation properties such as the variance of the gradient and preconditioner estimates. We note that our results also apply to subsampled Gauss-Newton algorithms for nonlinear least-squares problems, since these are recovered by setting $\mathcal{L}(v) = \frac{1}{2}\|v - b\|^2$.

## 3. Background

Recall that our formal starting point is the parametric optimization problem (1). To see the relevance of this problem, note that for NNWs we can set $v_\theta = \Psi_\theta / \|\Psi_\theta\|$ to be the normalized wavefunction and $\mathcal{L}(v) = \langle v, Hv \rangle$ to be the expectation value of the energy with respect to the Hamiltonian $H$. Stochastic minimization of the composite loss function $\mathcal{L}(v_\theta)$ then recovers the variational Monte Carlo method for finding ground states of quantum systems (Foulkes et al., 2001). Common PINN loss functions can also be written in the form of (1); see (Müller & Zeinhofer, 2023; 2024) for relevant treatments.

### 3.1. Natural Gradient Descent

From a geometric perspective, the intuition behind natural gradient descent is to mimic gradient descent in the function space (Müller & Zeinhofer, 2024). To describe this process we focus in on a single iteration denoted by index $t$ and parameters $\theta_t$, and we define the *function space gradient* $r = \nabla_v \mathcal{L}(v_{\theta_t}) \in \mathcal{H}$ and the *model Jacobian* $J = \nabla_\theta v_{\theta_t} : \mathbb{R}^n \to \mathcal{H}$. Here $r$ can be the standard

$L^2$ gradient or, for applications such as NNWs when the model $v_\theta$ is restricted to a manifold, a suitable Riemannian gradient. Given these definitions, an exact gradient step in the function space would satisfy $v_{\theta_{t+1}} = v_{\theta_t} - \eta r$ for some step size $\eta$. Applying the first-order approximation $v_{\theta_{t+1}} \approx v_{\theta_t} + J(\theta_{t+1} - \theta_t)$ and rearranging yields the *natural gradient subproblem*

$$J\theta_{t+1} = J\theta_t - \eta r. \tag{3}$$

In practice, the model Jacobian $J$ is not of full row rank and so the system must be solved by minimizing an appropriate function space distance metric. The most common approach is to use the $L^2$ distance and also incorporate Tikhonov regularization to yield the iteration

$$\theta_{t+1} = \arg\min_\theta \|J\theta - (J\theta_t - \eta r)\|^2 + \lambda \|\theta - \theta_t\|^2 \tag{4}$$

$$= \theta_t - \eta(J^\top J + \lambda I)^+ J^\top r, \tag{5}$$

where we have used the Moore-Penrose pseudoinverse to smoothly handle the case when $\lambda = 0$ and $J$ is not of full column rank. Here $J^\top J$ is known as the Fisher information matrix and the chain rule implies $J^\top r = \nabla_\theta \mathcal{L}(v_{\theta_t})$, so this formulation agrees with the standard preconditioner-based formulation of $L^2$ natural gradient descent. It also matches with the natural gradient algorithms that have been most effective for both NNWs and PINNs, which are commonly known as stochastic reconfiguration (Sorella, 2001) and energy natural gradient descent (Müller & Zeinhofer, 2023), respectively. See (Amari, 1998; Martens, 2020) for alternative discussions of the natural gradient method.

### 3.2. Sampling and Subsampling

To make natural gradient methods practical we can estimate both the function space gradient $r$ and the model Jacobian $J$ by sampling from the domain of $\mathcal{H}$. In particular we draw $k$ samples $S = (S_1, \ldots, S_k)$ with $S_i \in \Omega$, and we use these samples to instantiate

$$r_S = \begin{bmatrix} [\nabla_v \mathcal{L}(v_\theta)](S_1) \\ \vdots \\ [\nabla_v \mathcal{L}(v_\theta)](S_k) \end{bmatrix}, \quad J_S = \begin{bmatrix} \nabla_\theta^\top [v_\theta(S_1)] \\ \vdots \\ \nabla_\theta^\top [v_\theta(S_k)] \end{bmatrix}. \tag{6}$$

In practice the entries of $r_S$ can be computed efficiently as long as $\mathcal{L}$ has a local structure, while the rows of $J_S$ can be computed efficiently by backpropagating through the model. We note that when $S$ is sampled from a non-uniform distribution it may be appropriate to rescale the rows of $J_S, r_S$ via importance sampling, but for clarity we focus on the case without any rescaling. We also note that in the case of NNWs, both $r_S$ and $J_S$ include a normalization term which must be estimated stochastically. However, for our analysis we assume exact access to $r_S$ and $J_S$.

The *subsampled* regime refers to the case when the sample size $k$ is smaller than the number of parameters $n$, so that $J_S \in \mathbb{R}^{k \times n}$ is wider than it is tall. For example, a canonical application might use $k = 10^3$ samples to optimize $n = 10^6$ parameters. In such settings, substituting $J_S$ and $r_S$ into the regularized least-squares problem (4) yields an efficient subsampled natural gradient (SNG) algorithm:

$$\theta_{t+1} = \arg\min_\theta \|J_S\theta - (J_S\theta_t - \eta r_S)\|^2 + \lambda \|\theta - \theta_t\|^2$$

$$= \theta_t - \eta J_S^\top (J_S J_S^\top + \lambda I)^+ r_S$$

In addition to the efficiency gains from the sampling itself, this formulation only requires inverting a $k \times k$ kernel matrix in sample space rather than an $n \times n$ preconditioner matrix in parameter space, resulting in a dramatic reduction in the iteration cost.

To simplify the formulas we define the compressed notation

$$J_S^{+(\lambda)} = J_S^\top (J_S J_S^\top + \lambda I)^+, \qquad (7)$$

which should be viewed as a regularized version of the pseudoinverse since $J_S^{+(0)} = J_S^+$. This enables us to write the SNG iteration succinctly as

$$\theta_{t+1} = \theta_t - \eta J_S^{+(\lambda)} r_S. \qquad (8)$$

We note that although the values of $r, J, S$ change at each iteration, we keep this dependence implicit to avoid clutter.

### 3.3. Sketch-and-Project

Sketch-and-project methods solve a linear system $J\theta = b$ by repeatedly applying a sketching matrix $\Omega \in \mathbb{R}^{k \times m}$ on the left and then projecting the current iterate onto the solution space of the sketched equation $\Omega J\theta = \Omega b$ (Gower & Richtárik, 2015a). Applying this operation using standard orthogonal projections leads to the iterative method

$$\theta_{t+1} = \theta_t + (\Omega_t J)^+ (\Omega_t b - \Omega_t J\theta_t). \qquad (9)$$

In this work, we are most interested in the special case known as the randomized block Kaczmarz method (Needell & Tropp, 2014), which arises by choosing $\Omega_t = I_S$ to be some selected rows of the identity matrix, so that $\Omega_t J = J_S$ and $\Omega_t b = b_S$ represent the corresponding rows of $J$ and $b$. The resulting iteration takes the form

$$\theta_{t+1} = \theta_t + J_S^+ (b_S - J_S\theta_t). \qquad (10)$$

It is also possible to incorporate regularization into the randomized block Kaczmarz iterations to yield the formula

$$\theta_{t+1} = \theta_t + J_S^{+(\lambda)} (b_S - J_S\theta_t). \qquad (11)$$

Such regularization has been shown to yield improved stability for inconsistent linear systems (Goldshlager et al., 2025)

and to enable optimal convergence rates in the presence of Nesterov acceleration (Dereziński et al., 2025b). We emphasize that the regularization here is internal to the iterations and does *not* imply that the algorithm targets a regularized solution to the original linear system.

The standard convergence result for sketch-and-project methods, for example Theorem 1.1 of (Gower & Richtárik, 2015b), implies that when $\Omega_t = I_S$ and when there is a unique $\theta^*$ satisfying $J\theta^* = b$, it holds

$$\mathbb{E} \|\theta_t - \theta^*\|^2 \le (1 - \alpha)^t \|\theta_0 - \theta^*\|^2, \qquad (12)$$

with

$$\alpha = \lambda_{\min}^+(\overline{P}), \ \overline{P} = \mathbb{E}_S[P(S)], \ P(S) = J_S^{+(\lambda)} J_S. \qquad (13)$$

In words, $P(S)$ is a regularized projector onto the row space of $J_S$, $\overline{P}$ is the expectation of this projector, and $\alpha$ is the minimal eigenvalue of the expected projector.

A major strength of sketch-and-project methods is that their convergence rate can be insensitive to the largest $k$ singular values of $J$, where we recall that $k$ is the sketch size or in the case of randomized block Kaczmarz methods, the sample size. As a result, the rate constant $\alpha$ can scale superlinearly in the sample size in the presence of fast spectral decay. In fact, under fast enough spectral decay and other appropriate conditions, $\alpha$ can scale as a high-order polynomial or even exponentially in $k$; see Appendix B for details.

### 3.4. Squared Volume Sampling

Given a matrix $J$, a sample size $k$, and a regularization parameter $\lambda \ge 0$, squared volume sampling (SVS) selects $k$ rows from $J$ with indices $S$ with a probability proportional to the squared volume of the parallelepiped formed by the rows of $[J_S \ \sqrt{\lambda}I]$. Concretely we define the distribution $\text{SVS}(J, k, \lambda)$ to have probability mass function

$$p(S) = \frac{\det(J_S J_S^\top + \lambda I)}{\sum_{|S'|=k} \det(J_{S'} J_{S'}^\top + \lambda I)} \qquad (14)$$

when $|S| = k$, and 0 otherwise. We note that when $\lambda = 0$ it is required that $k \le \text{rank}(J)$, and we also remark that SVS is a special case of a determinantal point process (DPP).

Various algorithms have been proposed to implement squared volume sampling, including both direct and Markov chain Monte Carlo approaches; see (Derezinski & Mahoney, 2021) and (Calandriello et al., 2020) for relevant discussions. However, our main interest in SVS is not as a practical sampling strategy, but rather as a tool to enable theoretical insight into algorithms which are otherwise difficult to analyze. SVS has previously been used in this way for both randomized block coordinate descent methods (Mutny et al., 2020; Rodomanov & Kropotov, 2020) and randomized block Kaczmarz methods (Dereziński et al., 2025b;

Goldshlager et al., 2025; Xiang et al., 2025). In both cases, the power of SVS is that it enables exact computations of expectation values involving the inverse of a subsampled matrix, which are otherwise challenging to control. We will use this same property to provide new insights into subsampled natural gradient algorithms.

### 3.5. Related Works

Several works have made connections between subsampled Newton algorithms and sketch-and-project methods in both linear (Rathore et al., 2024) and nonlinear (Qu et al., 2016; Yuan et al., 2022) settings, with one even considering a variant of squared volume sampling (Mutny et al., 2020). Another work has analyzed a subsampled Gauss-Newton algorithm for overparameterized neural networks in a nonlinear least-squares setting (Cai et al., 2019). In the context of natural gradient methods, one recent work considered the use of squared volume sampling (Gruhlke et al., 2024), but this work focused on the oversampled regime in which there are more samples than parameters, which is qualitatively different than the subsampled regime.

## 4. Analysis without Decoupling

We now begin our work by considering the primary barrier to providing theoretical insight using the sketch-and-project approach: namely, how to rigorously guarantee the convergence without decoupling the gradient from the preconditioner. To start, we note that the SNG iteration (8) can be derived by applying a regularized block Kaczmarz step (11) to the natural gradient subproblem (3):

$$\theta_{t+1} = \theta_t + J_S^{+(\lambda)}((J\theta_t - \eta r)_S - J_S\theta_t)$$
$$= \theta_t - \eta J_S^{+(\lambda)} r_S.$$

This forms the basis for the sketch-and-project interpretation of SNG, and it implies rigorous convergence guarantees such as (12) for consistent linear least-squares problems. However, it is not trivial to extend such guarantees to more general settings when both the left- and right-hand side of the linear system are changing at each iteration.

To clarify the problem we write the expected direction as

$$\mathbb{E}\left[J_S^{+(\lambda)} r_S\right] = J^\top \overline{W} r \tag{15}$$

where $\overline{W} = \mathbb{E}_S\left[I_S^\top (J_S J_S^\top + \lambda I)^+ I_S\right] \succeq 0$. To guarantee the convergence using standard techniques, we would like to pass $\overline{W}$ to the left as $J^\top \overline{W} r = \widetilde{W} J^\top r$, for some appropriate $\widetilde{W} \succ 0$, to obtain a preconditioned gradient descent direction. Unfortunately it is difficult to guarantee that this manipulation is possible under general conditions.

In fact, it is exactly this challenge that has led previous analyses to study a theoretical proxy based on two independent

mini-batches. However, as we have alluded to and as we discuss in detail in Appendix A, such analyses are not able to provide meaningful characterizations of the convergence rate of SNG at small sample sizes. Motivated by this failure, we return to the single mini-batch algorithm and search for another way to facilitate the analysis. In particular, the interpretation of SNG as a randomized block Kaczmarz method suggests the consideration of a new proxy based on squared volume sampling (SVS). This proxy is promising since SVS is known to enable exact computations of expectation values along the lines of (15), and we find that it is indeed sufficient to address the challenge at hand:

**Lemma 4.1.** *Let $J \in \mathbb{R}^{m \times n}$, $r \in \mathbb{R}^m$, and $S \sim SVS(J, k, \lambda)$. Then it holds*

$$\mathbb{E}_S\left[J_S^{+(\lambda)} r_S\right] = \widetilde{W} J^\top r, \tag{16}$$

*with $\widetilde{W} = (J^\top J)^{-1/2} \overline{P} (J^\top J)^{-1/2}$ and $\overline{P}$ as in (13).*

In other words, under SVS the expectation of the SNG direction $J_S^{+(\lambda)} r_S$ becomes equal to a preconditioned gradient descent step, *without* decoupling the gradient and the preconditioner. Moreover, the SNG preconditioner $\widetilde{W}$ arises by combining the expected projector $\overline{P}$ with the natural gradient preconditioner $J^\top J$, which already suggests that the sketch-and-project structure controls the gap between SNG and its deterministic counterpart. The proof of Lemma 4.1, which follows from known expectation formulas for SVS, can be found in Appendix C. Note that here and elsewhere we use $(J^\top J)^{-1/2}$ as a shorthand for $[(J^\top J)^+]^{1/2}$.

As a first consequence of Lemma 4.1, we can now provide global convergence guarantees for SNG without invoking two independent mini-batches. The assumptions to ensure global convergence can be formulated in a variety of ways; to provide just one example we present the following result:

**Theorem 4.2** (Global convergence of SVS-SNG)**.** *Consider an instance of the parametric optimization problem (1) such that $\mathcal{H} = \mathbb{R}^m$. Suppose that $\mathcal{L}(v_\theta)$ is bounded below and is continuously differentiable as a function of $\theta$, and also that $v_\theta, \nabla_\theta \mathcal{L}(v_\theta)$ are Lipschitz continuous as a function of $\theta$. Suppose additionally that there exist constants $C_1, C_2 > 0$ such that $\|r\|^2 \leq C_1 \|J^\top r\|^2 + C_2$ for all $\theta$.*

*Then under the sampling distribution $S \sim SVS(J, k, \lambda)$, a positive regularization $\lambda > 0$, and diminishing step sizes $\eta_t$ satisfying $\sum_{t=1}^\infty \eta_t = \infty$ and $\sum_{t=1}^\infty \eta_t^2 < \infty$, it holds*

$$\lim_{T \to \infty} \mathbb{E}\left[\frac{1}{\sum_{t=1}^T \eta_t} \sum_{t=1}^T \eta_t \|\nabla_\theta \mathcal{L}(v_{\theta_t})\|^2\right] = 0.$$

The only unusual part of this result is the condition on $\|r\|^2$, which is used to ensure that the variance of the sketch-and-project step can be controlled in terms of the size of

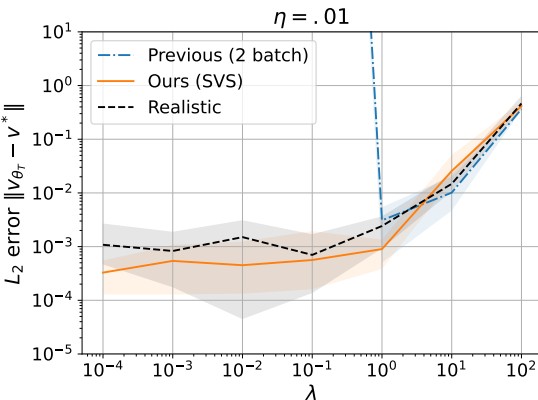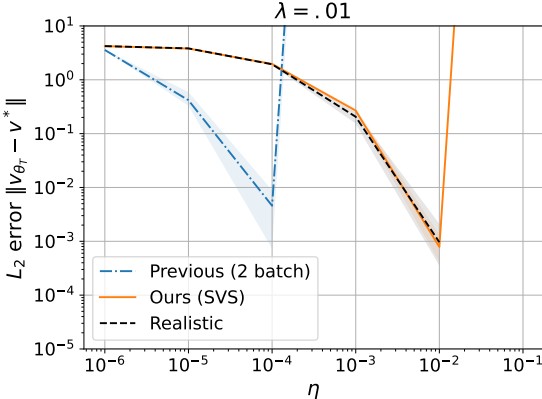

*Figure 2.* Empirical performance of two theoretical proxies for SNG. "Previous (2 batch)" refers to (25) with $S, S'$ uniform, "Ours (SVS)" refers to (8) with squared volume sampling, and "Realistic" refers to (8) with $S$ uniform. The results are for a discrete Poisson problem with a small neural network ansatz, and the behavior is tested for a fixed step size $\eta$ and varying regularization $\lambda$ (left), as well as for fixed $\lambda$ and varying $\eta$ (right). In both cases the behavior of SVS-SNG matches closely with the realistic algorithm, while the behavior of the two mini-batch proxy differs dramatically. The dimensions are $m = 100$, $n = 7801$, and $k = 10$, which is small enough to directly implement SVS. The $y$-axis represents the relative error achieved after $T = 10^3$ iterations, with the curves representing the mean result over five runs and the shaded regions spanning the minimal and maximal results attained over the same five runs. See also Appendix I for similar results in the linear least-quadratics setting, and Appendix J for full details on these and other numerical experiments, including a link to the code which can be used to reproduce our results.

the gradient $J^\top r$. To see that this condition is reasonable, we note every LLQ instance admits values of $C_1$ and $C_2$ such that the condition holds, as proved in Proposition D.2. The proof of Theorem 4.2 is also presented in Appendix D and follows by showing that under SVS, SNG satisfies the assumptions of Theorem 4.10 of (Bottou et al., 2018). We emphasize that SVS is merely a *sufficient* condition for these assumptions, and it is possible that other sampling distributions can also satisfy these assumptions.

The main value of Lemma 4.1 and Theorem 4.2 is in establishing SVS-SNG as a *new theoretical proxy* for understanding SNG. The suitability of SVS as a proxy is supported by previous analyses of randomized block Kaczmarz methods which show that under benign conditions, SVS can behave similarly to uniform sampling (Dereziński et al., 2025b; Goldshlager et al., 2025). We further provide numerical evidence that this similarity can extend to the case of SNG, with results for a simple scientific machine learning problem in Figure 2 and for some linear least-quadratics problems in Appendix I. See Appendix J for full details on these and other numerical experiments, including a link to the code which can be used to reproduce our results.

Regarding the practical implications, we do not suggest that SVS should be implemented directly, since even the most efficient algorithms for doing so still require accessing either all of $J$ or a large number of rows of $J$ in order to generate each mini-batch. However, the idea of SVS can still motivate new practical directions, for example by suggesting the incorporation of negative correlations between the

samples within each mini-batch. It may even be possible to develop cheaper, approximate adaptations of rigorous SVS algorithms such as the Markov chain Monte Carlo method of (Anari et al., 2016) or the oversample-then-subselect approach of (Calandriello et al., 2020). These methods have complexities like $\mathcal{O}(m \cdot \mathrm{poly}(k))$ in general, but the linear scaling in $m$ could be reduced by running fewer MCMC steps or using a smaller oversampling factor, respectively.

## 5. Small-Sample Insights

We now turn our attention to our main goal of providing detailed insights into small-sample settings. For this purpose we focus on the linear least-quadratics setting (LLQ), which we recall is motivated by the search for high-precision solutions and is derived as a second-order approximation to general consistent parametric optimization problems. For clarity and since the motivation for the model problem comes from the consistent regime, we focus on consistent instances of LLQ. However, in Appendix G we provide an extension to the more general case when $\mathcal{L}$ is smooth and strongly convex and the problem is possibly inconsistent. We also note that throughout this section we uniquely define $\theta^* = \arg\min_{\theta \in \mathrm{range}(J^\top)} \mathcal{L}(v_\theta)$.

Within the consistent LLQ setting, the key challenge lies in handling the interaction between (i) the sketch-and-project structure induced by the Jacobian $J$, and (ii) the natural gradient structure induced by the Hessian $H$. Fortunately, we already know from Lemma 4.1 that under SVS, the expectation of each SNG iteration takes the form of a precondi-

tioned gradient descent step. This enables us to extend existing sketch-and-project analyses to ensure a steady reduction in the solution error when measured in an appropriate norm:

**Theorem 5.1** (Convergence of SVS-SNG for LLQ). *Consider a consistent instance of linear least-quadratics and assume an initial guess of $\theta_0 \in \mathrm{range}(J^\top)$. Define $\overline{P}, \alpha$ as in (13), $\widetilde{W}$ as in Lemma 4.1, and a new quantity*

$$\gamma = \left\| \mathbb{E}_S \left[ I_S^\top (J_S^{+(\lambda)})^\top \widetilde{W}^+ J_S^{+(\lambda)} I_S \right] \right\|_2 \qquad (17)$$

*which relates to the second moment of the sketch-and-project step.*

*Then when $S \sim SVS(J, k, \lambda)$ and $\eta \leq 1/(\gamma \|H\|_2)$, it holds*

$$\mathbb{E} \|\theta_t - \theta^*\|_{\widetilde{W}^+}^2 \leq (1 - \eta \cdot \lambda_{\min}(H) \cdot \alpha)^t \|\theta_0 - \theta^*\|_{\widetilde{W}^+}^2 . \qquad (18)$$

The proof of Theorem 5.1 can be found in Appendix E. Importantly, although $\widetilde{W}^+$ is not necessarily full rank, it does act as a norm on the subspace $\mathrm{range}(J^\top)$, which contains all the error vectors as established by Lemma E.1. As a result, Theorem 5.1 provides a genuine linear convergence rate for SVS-SNG for LLQ. We note that unlike Theorem 4.2, which ultimately relies on inequality conditions that might be met by more general sampling distributions, Theorem 5.1 directly leverages the exact expectation formula of Lemma 4.1 and can thus be more difficult to generalize.

To understand the meaning of Theorem 5.1 it is helpful to compare with deterministic natural gradient descent:

**Corollary 5.2** (Convergence of NGD for LLQ). *Consider a consistent instance of linear least-quadratics and assume an initial guess of $\theta_0 \in \mathrm{range}(J^\top)$. Then when $\lambda = 0$ and $\eta \leq 1/\|H\|_2$, deterministic natural gradient descent satisfies*

$$\mathbb{E} \|\theta_t - \theta^*\|_{J^\top J}^2 \leq (1 - \eta \cdot \lambda_{\min}(H))^t \|\theta_0 - \theta^*\|_{J^\top J}^2 . \qquad (19)$$

The proof of Corollary 5.2, which is a direct specialization of Theorem 5.1, can be found in Appendix E.

Overall, these results provide the first characterization of the convergence properties of SNG in small-sample settings. In particular, comparing Corollary 5.2 to Theorem 5.1 reveals that under SVS, the effect of subsampling is to (i) reduce the maximum stable step size by a factor of $\gamma$, and (ii) reduce the convergence rate by an additional factor of $\alpha$. This means that as the sample size is varied, the optimal rate scales as $\alpha/\gamma$.

Importantly, the appearance of $\alpha$ in Theorem 5.1 enables us to draw insights from the rich body of literature on sketch-and-project methods. For example, based on this literature we can immediately suggest a new explanation for the success of SNG in scientific machine learning, since (i) it is

known that $\alpha$ can scale superlinearly in the sample size when $J$ exhibits fast spectral decay (see Section 3.3 and Appendix B), and (ii) $J$ is commonly observed to exhibit fast spectral decay in practice (Park & Kastoryano, 2020; Wang et al., 2022). This provides a distinct advantage relative to SGD, for which the analogous linear convergence rate constant for smooth and strongly convex problems scales at most linearly in the sample size across a variety of standard settings (Jain et al., 2018; Mishkin, 2020; Garrigos & Gower, 2023). Furthermore, since the scaling of $\alpha$ is limited only by the spectral decay rate, this superlinear scaling can easily compensate for SNG's increased iteration cost of $\mathcal{O}(k^2)$ rather than $\mathcal{O}(k)$. It is also worth noting that the superlinear scaling of $\alpha$ can occur independently of any changes to the step size $\eta$, which depends only on $H$ and $\gamma$. This again contrasts with SGD, for which the step size must always be increased in order to attain a faster linear convergence rate.

The other sketch-and-project quantity, $\gamma$, has not appeared in previous analyses. Intuitively, the effect of $\gamma$ is to modulate the benefits of the sketch-and-project structure, since when $\gamma = \mathcal{O}(1)$ the sketch-and-project rate $\alpha$ extends cleanly into the natural gradient setting, whereas when $\gamma \gg 1$ the rate suffers from the transition. Structurally, $\gamma$ is related to the second moment of the sketch-and-project step and is similar to second-moment quantities which have appeared in analyses of accelerated sketch-and-project methods (Gower et al., 2018; Dereziński et al., 2025b). As discussed in Section 6, such second-moment quantities can admit favorable upper bounds in benign cases, which suggests that $\gamma$ may also behave benignly under appropriate conditions.

Unfortunately, proving such a result can be quite technical and is outside the scope of the current work. For now we present the following preliminary characterizations:

**Proposition 5.3.** *Define $\gamma$ as in Theorem 5.1. Then:*

1. *If $JJ^\top = I$, then uniform and squared volume sampling coincide, and under such sampling $\gamma = \frac{1}{1+\lambda}$.*

2. *If $k = 1$ and $S \sim SVS(J, k, \lambda)$, then $\gamma \leq 1$.*

3. *If $k = m$, then $\gamma \leq 1$.*

4. *If $\lambda > 0$ and $S$ is uniform, then $\gamma \leq 1 + \frac{k}{\lambda} \max_i \|J_i\|^2$.*

The proof of Proposition 5.3 is presented in Appendix E. These results put some limits on the extent to which $\gamma$ can suppress the superlinear scaling enjoyed by $\alpha$, but they are not tight enough to fully clarify the behavior. To supplement these preliminary bounds, we provide an additional proposition in Appendix F which shows that the influence of $\gamma$ can disappear entirely when $J$ and $H$ satisfy a strong compatibility condition. We also provide a numerical experiment which explores the behavior of $\alpha$ and $\gamma$ under SVS,

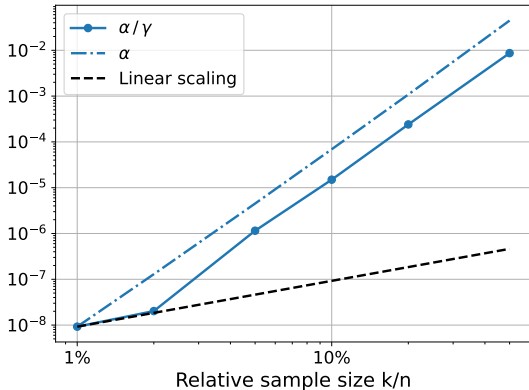

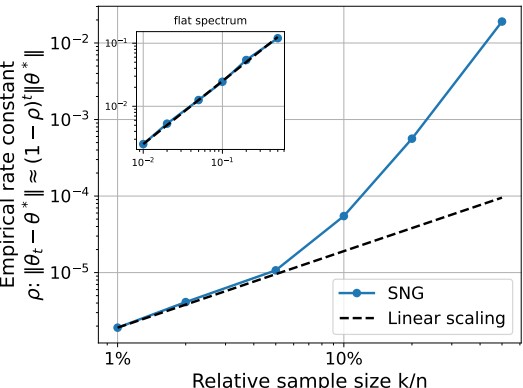

*Figure 3.* Empirical scaling of the convergence parameters $\alpha$ and $\gamma$ as a function of the sample size $k$, under SVS. The results are for a random Gaussian Jacobian matrix with quadratic spectral decay. The sketch-and-project constant $\alpha$ grows superlinearly as predicted, and the presence of $\gamma$ only mildly impedes this superlinear scaling. The regularization is set to $\lambda = 0$ for simplicity and the dimensions are $m = 10^3$ and $n = 10^2$, which is small enough to (i) calculate $\alpha$ directly using symmetric polynomials, and (ii) estimate $\gamma$ by sampling directly from SVS($J, k, \lambda$).

*Figure 4.* Empirical validation that SNG can exploit spectral decay in the model Jacobian, under uniform sampling. The results are for a consistent instance of linear least-quadratics involving random Gaussian matrices, with $J$ exhibiting quadratic spectral decay. The empirical rate constant grows superlinearly as predicted even while the step size is fixed at $\eta = 1$. The regularization is set to $\lambda = 0$ for simplicity and the dimensions are $m = 10^3$ and $n = 10^2$. Inset: when $J$ has a flat spectrum, the rate scales linearly.

for a random Gaussian $J$ matrix with quadratic spectral decay. The results in Figure 3 show that $\gamma$ exerts only a mild damping effect on the predicted convergence rate, with $\alpha/\gamma$ still growing superlinearly.

Finally, we directly investigate the suggestion that SNG can exploit spectral decay in the model Jacobian. To this end we test the behavior of SNG under uniform sampling for two instances of LLQ involving random Gaussian matrices. The results in Figure 4 confirm that SNG exhibits superlinear scaling exactly when $J$ has fast spectral decay.

## 6. Principled Acceleration

As an additional demonstration of the power of the sketch-and-project approach, we now turn our attention to the question of acceleration. In particular, we consider the subsampled projected-increment natural gradient (SPRING) algorithm, which has recently been proposed as a structured momentum scheme for SNG, and which has been demonstrated to significantly accelerate the convergence in practice (Goldshlager et al., 2024). Despite numerous applications of SPRING (Smith et al., 2024; Scherbela et al., 2025; Chen et al., 2025; Hendry et al., 2025) and several proposals to further improve the algorithm (Gu et al., 2025; Li et al., 2025), the structure and convergence properties of SPRING have remained mysterious. We clarify the situation through the following result:

**Theorem 6.1** (SPRING as accelerated sketch-and-project).
*The Nesterov-accelerated regularized sketch-and-project algorithm for the system $J\theta = b$, as presented for example*

*by (Gower et al., 2018; Dereziński et al., 2025b), can be reformulated as*

$$\phi_{t+1} = \mu\phi_t + (\Omega_t J)^{+(\lambda)}(\Omega_t b - \Omega_t J(\theta_t + \mu\phi_t)),$$
$$\theta_{t+1} = \theta_t + \eta\phi_{t+1} \tag{20}$$

*after an appropriate transformation of variables. Applying (20) to the natural gradient subproblem (3), with the standard row sampling sketching matrix $I_S$, yields the iteration*

$$\phi_{t+1} = \mu\phi_t + J_S^{+(\lambda)}(r_S - \mu J_S\phi_t),$$
$$\theta_{t+1} = \theta_t - \eta\phi_{t+1}. \tag{21}$$

*This is exactly the iteration used by SPRING.*

In other words, while the SPRING algorithm appears mysterious from a stochastic preconditioning perspective, it is a natural construction from the viewpoint of sketch-and-project methods. The proof of Theorem 6.1, which is straightforward once the appropriate transformation of variables is identified, can be found in Appendix H.

As a result of Theorem 6.1, Theorem 3 of (Gower et al., 2018) (or also eq. (1) of (Dereziński et al., 2025a)) implies that for a consistent linear least-squares problem SPRING satisfies a convergence bound of the form

$$\mathbb{E}\,\|\theta_t - \theta^*\|^2 = \mathcal{O}\left((1 - \sqrt{\alpha/\beta})^t\right)\|\theta_0 - \theta^*\|^2 \tag{22}$$

with $\alpha$ as in (13), $\mathcal{O}$ indicating constant factors that do not depend on $t$, and $\beta$ a second moment quantity defined by

$$\beta = \left\|\overline{P}^{-1/2}\mathbb{E}_S\left[J_S^\top(J_S^{+(\lambda)})^\top\overline{P}^+J_S^{+(\lambda)}J_S\right]\overline{P}^{-1/2}\right\|_2. \tag{23}$$

The implications of this bound have been developed by several previous works (Gower et al., 2018; Dereziński et al., 2025a;b), with the overall message being that the rate constant $\sqrt{\alpha/\beta}$ can be similar to $\alpha$ for adversarial problems, but can scale more like $\sqrt{\alpha}$ for benign problems. Conceptually, this means that we can expect the benefits of SPRING to be greatest when $\alpha$ is small, or equivalently when the underlying sketch-and-project steps converge slowly. This explains the recent observations of (Guzmán-Cordero et al., 2025; Misery et al., 2026) that SPRING is most beneficial when the sample size is small relative to the difficulty of the problem. Finally, we note that the benign case follows from providing favorable upper bounds on $\beta$, which suggests that similar techniques could also be used to bound our $\gamma$.

We are not yet able to extend these results to the LLQ setting, since it is unclear how to construct an appropriate Lyapunov function in the presence of both Nesterov acceleration and a nontrivial function space Hessian. Still, based on Theorems 5.1 and 6.1 and eqs. (12) and (22), we conjecture that SPRING always accelerates SNG by replacing $\alpha$ with the accelerated sketch-and-project constant $\sqrt{\alpha/\beta}$:

**Conjecture 6.2** (Convergence of SVS-SPRING for LLQ).
*Consider a consistent instance of linear least-quadratics and assume an initial guess of $\theta_0 \in \mathrm{range}(J^\top)$. Then under the sampling distribution SVS($J, k, \lambda$) and with appropriate choices of $\eta$ and $\mu$, SPRING satisfies*

$$\mathbb{E} \, \|\theta_t - \theta^*\|^2 = \mathcal{O}\left( \left(1 - \kappa^{-1}(H) \cdot \sqrt{\alpha/\beta} \, / \, \gamma\right)^t \right) \|\theta_0 - \theta^*\|^2 \tag{24}$$

*with $\alpha$ as in (13), $\beta$ as in (23), $\gamma$ as in Theorem 5.1, and $\mathcal{O}$ indicating constant factors that do not depend on $t$.*

We defer the resolution of this conjecture to future works. For now, we provide numerical experiments exploring the convergence of SPRING in the LLQ setting, with results in Figure 5. We find that SPRING provides substantial acceleration for small sample sizes, but the gap shrinks for larger sample sizes. This is consistent with Conjecture 6.2 since large sample sizes correspond exactly to larger values of $\alpha$, which should thus reduce the opportunity for acceleration.

Interestingly, we do *not* predict or observe SPRING to achieve an improvement in the dependence on the condition number $\kappa(H)$. Identifying algorithms that do, perhaps achieving convergence rates scaling with $\kappa^{-1/2}(H)$, for example by introducing acceleration in the function space (Li et al., 2025), is a promising direction for future works.

## Conclusions

In this work we have demonstrated that subsampled natural gradient algorithms can be more fruitfully understood as sketch-and-project methods rather than stochastic preconditioning schemes. Building on this perspective we have

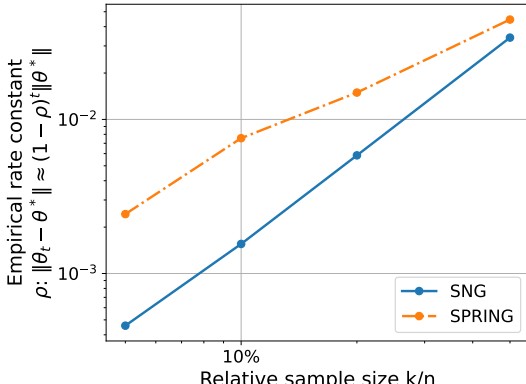

*Figure 5.* Empirical comparison of the convergence rates of SNG and SPRING for a variety of sample sizes. The results are for a consistent instance of linear least-quadratics in which $J$ and $H$ both have linearly decaying spectra. As suggested by Conjecture 6.2, SPRING provides substantial acceleration for small sample sizes, but the gap shrinks as the sample size is increased. The regularization is set to $\lambda = 0$ for simplicity and the dimensions are $m = 10^3$, $n = 10^2$. Furthermore, the hyperparameters $\eta$ and $\mu$ are tuned independently for each run.

shown how to analyze SNG without decoupling the gradient and the preconditioner, provided the first meaningful characterization of how subsampling affects the convergence rate of natural gradient descent, and clarified the mechanism behind the SPRING algorithm. Overall, our findings suggest that when using subsampled natural gradient algorithms to solve scientific machine learning problems, more attention should be paid to the sketch-and-project properties such as $\alpha$ and $\gamma$, and less to the statistical estimation properties such as the variance of the gradient and preconditioner estimates.

Looking forward, our work sets the stage for a broader effort to leverage ideas from randomized linear algebra to develop and analyze subsampled optimizers for scientific machine learning. Theoretically, promising directions include providing tighter bounds on $\gamma$, extending our analysis to other sampling distributions, and generalizing our convergence rate characterization beyond the case of a linear ansatz. Regarding acceleration it would be interesting to verify Conjecture 6.2 and to identify schemes that attain convergence rates scaling with $\kappa^{-1/2}(H)$. Practically, future works should explore sampling algorithms inspired by SVS, for example by using ensemble samplers (Goodman & Weare, 2010) to introduce negative correlation within each mini-batch, or by proposing practical adaptations of the methods of (Anari et al., 2016) or (Calandriello et al., 2020). It should also be fruitful to explore additional tools from the sketch-and-project literature such as adaptive acceleration (Dereziński et al., 2025b), tail averaging (Epperly et al., 2025), and oblique projections (Gower & Richtárik, 2015a), which could lead to faster and more robust convergence.

# Acknowledgements

This material is based upon work supported by the U.S. Department of Energy, Office of Science, Office of Advanced Scientific Computing Research, Department of Energy Computational Science Graduate Fellowship under Award Number DE-SC0023112 (G.G.). This work was supported by a grant from the Simons Foundation [SFI-MPS-SDF-00014421, G.G.] and by a Hearts to Humanity Eternal graduate research grant (G.G.). This work was supported in part by the U.S. Department of Energy, Office of Science, Office of Advanced Scientific Computing Research's Applied Mathematics Competitive Portfolios program under Contract No. AC02-05CH11231 (L.L.), and by the Simons Investigator in Mathematics award through Grant No. 825053 (J.H., L.L.). We thank Ethan Epperly, James Larsen, Michal Dereziński, and Marius Zeinhofer for helpful discussions.

# Impact statement

This paper presents work whose goal is to advance the field of scientific machine learning, with a focus on understanding and improving optimization algorithms for training neural networks to solve quantum many-body problems and partial differential equations. There are many potential societal impacts of advancements in these application areas, both positive and negative, none of which we feel must be specifically highlighted here.

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

## A. Detailed Discussion of the Two Mini-Batch Proxy

In this appendix we provide a more detailed discussion of the two mini-batch proxy which has been analyzed by previous works such as (Ren & Goldfarb, 2019; Martens, 2020; Yang et al., 2020; Wu et al., 2024), and which is similar to the algorithms analyzed by works such as (Bollapragada et al., 2019; Roosta-Khorasani & Mahoney, 2019) in the context of subsampled Newton methods. The basic idea is to study an algorithm of the form

$$\theta_{t+1} = \theta_t - \eta(J_{S'}^\top J_{S'} + \lambda I)^+ J_S^\top r_S \tag{25}$$

where $S, S'$ are two independent and uniformly sampled mini-batches. This form is justified as a plug-in estimator for the deterministic algorithm (5), and when $S = S'$ it becomes equivalent to the practical single mini-batch algorithm (8) due to the push-through identity $(J_S^\top J_S + \lambda I)^+ J_S^\top = J_S^\top (J_S J_S^\top + \lambda I)^+$.

The benefit of studying (25) is that its convergence can be guaranteed under mild conditions since its preconditioner $J_{S'}^\top J_{S'} + \lambda I$ is positive definite (when $\lambda > 0$) and independent from the stochastic gradient $J_S^\top r_S$, which is an unbiased estimator for the exact gradient $\nabla_\theta \mathcal{L}(v_\theta) = J^\top r$ up to a scaling factor of $k/m$. The problem is that at small sample sizes the meaningful part of the preconditioner, $J_{S'}^\top J_{S'}$, has a rank of at most $k \ll n$ and is thus unlikely to be informative for the independently estimated stochastic gradient $J_S^\top r_S$. As a result of this problem, analyses of two mini-batch algorithms such as (25) are not able to provide meaningful characterizations of the convergence rate of SNG at small sample sizes.

## B. Detailed Discussion of Sketch-and-Project Convergence Rates

In this appendix we provide a more detailed discussion of the scaling properties of the convergence parameter $\alpha$ from (12). A representative result is eq. (4.2) of (Dereziński & Yang, 2024). Assuming that $J$ is of full column rank and that $S \sim \text{SVS}(J, k, 0)$, this equation provides a lower bound of

$$\alpha \geq \max_{1 \leq \ell < k} \frac{k - \ell}{k - \ell - 1 + \sum_{i=\ell+1}^n (\sigma_i/\sigma_n)^2}, \tag{26}$$

where $\sigma_i$ represent the singular values of $J$. Fixing $\ell = k - 1$ provides a simpler bound of

$$\alpha \geq \frac{\sigma_n^2}{\sum_{i=k}^n \sigma_i^2}. \tag{27}$$

Evidently, the right-hand side of (27) can scale rapidly with $k$ when $J$ exhibits fast spectral decay.

For concrete results along these lines under various types of sketching and spectral decay, see Lemma 3.3 and Corollary 3.4 of (Dereziński & Rebrova, 2024). These results show that $\alpha$ can scale as a high-order polynomial or even an exponential function in $k$, when the spectral decay of $J$ is similar. Additionally, for a discussion of how regularization impacts the scaling of $\alpha$, see Theorem E.1 and the associated discussion in (Goldshlager et al., 2025).

## C. Proof of Lemma 4.1

We first reproduce a known result regarding the expectation of the subsampled projection matrix under squared volume sampling:

**Lemma C.1** (Key expectation formula under SVS). *Let $J \in \mathbb{R}^{m \times n}$ with rank $e$ and full SVD $\mathbf{U}\boldsymbol{\Sigma}\mathbf{V}^\top$, and suppose $S \sim SVS(J, k, \lambda)$. Then*

$$\overline{W} = \mathbb{E}_S \left[ I_S^\top (J_S J_S^\top + \lambda I)^+ I_S \right] = \mathbf{U}\mathbf{D}\mathbf{U}^\top \tag{28}$$

*where*

$$\mathbf{D} = \frac{\text{diag}(p_{k-1}(q_{-1}), \ldots, p_{k-1}(q_{-m}))}{p_k(q)}, \tag{29}$$

*with $q$ the vector of descending eigenvalues of $JJ^\top + \lambda I$ ($q_i = \sigma_i^2 + \lambda$ when $i \leq e$, $q_i = \lambda$ when $i > e$), $q_{-i}$ the vector $q$ with the $i^{th}$ entry removed, and $p_i$ the real elementary symmetric polynomial of degree $i$.*

*Proof.* At a high level this result is proved by applying the identity $X^{-1} = \frac{1}{\det(X)}\text{adj}(X)$ with $X = J_S J_S^\top + \lambda I$, then noting that the squared volume sampling probability density cancels with the determinant. What remains is essentially a

sum over the adjugates of the principal submatrices of $JJ^\top + \lambda I$, which turns out to have an elegant formula in terms of symmetric polynomials of the eigenvalues, yielding the numerator of (29). The normalization factor of the squared volume sampling density also has such a formula, which yields the denominator of (29).

For the full details of the proof, including how to handle the case when $J_S J_S^\top + \lambda I$ is singular (which can happen when $\lambda = 0$), see the proof of Lemma 4.1 in (Dereziński & Yang, 2024). In particular, their eq. (5.2) is equivalent to our Lemma C.1. See also (Rodomanov & Kropotov, 2020), whose Lemmas 3.2 and 3.3 provide the foundation for the results of (Dereziński & Yang, 2024). □

We now present a corollary of Lemma C.1 which will be useful in the proof of Lemma 4.1 and elsewhere:

**Corollary C.2** (Alternative characterization of $\widetilde{W}$). *Let $e$ be the rank of $J$ and let $U\Sigma V^\top$ be its compact SVD. Fix $k$ and $\lambda$ and define $\widetilde{W} = (J^\top J)^{-1/2}\overline{P}(J^\top J)^{-1/2}$ as in Lemma 4.1. Then it holds*

$$\widetilde{W} = VDV^\top \tag{30}$$

*where $D = \mathbf{D}[1:e, 1:e]$ represents the $e^{th}$ leading principal submatrix of $\mathbf{D}$, which is defined in (29). It also holds $\lambda_{\min}(D) \geq 1/\operatorname{Tr}(JJ^\top + \lambda I)$.*

*Proof.* First note

$$\overline{P} = \mathbb{E}_S\left[J_S^{+(\lambda)}J_S\right] = J^\top \mathbb{E}_S\left[I_S^\top(J_S J_S^\top + \lambda I)^+ I_S\right]J = J^\top \overline{W}J. \tag{31}$$

Then substitute

$$\widetilde{W} = (J^\top J)^{-1/2}\overline{P}(J^\top J)^{-1/2} = (J^\top J)^{-1/2}J^\top\overline{W}J(J^\top J)^{-1/2} = VU^\top(\mathbf{UDU}^\top)UV^\top = VDV^\top. \tag{32}$$

Finally, using eq. (5.3) of (Dereziński & Yang, 2024) we can lower bound

$$\lambda_{\min}(D) \geq \min_i \frac{p_{k-1}(q_{-i})}{p_k(q)} \geq \min_i \frac{1}{q_i + \frac{1}{k-1}\sum_{j=2, j\neq i}^m q_j} \geq \frac{1}{q_1 + \frac{1}{k-1}\sum_{j=2}^m q_j} \geq \frac{1}{\sum_{j=1}^m q_j} = \frac{1}{\operatorname{Tr}(JJ^\top + \lambda I)}. \tag{33}$$

This chain of inequalities is ill-defined when $k = 1$, but it is easy to verify that in this case, $p_{k-1}(q_{-i})/p_k(q) = 1/\operatorname{Tr}(JJ^\top + \lambda I)$ for all $i$ and so the bound is still valid. □

We are now ready to prove Lemma 4.1:

*Proof of Lemma 4.1.* Let $e$ be the rank of $J$ and let $U\Sigma V^\top$ be its compact SVD. Calculate

$$\begin{aligned}
\mathbb{E}_S\left[J_S^{+(\lambda)}r_S\right] &= \mathbb{E}_S\left[J_S^\top(J_S J_S^\top + \lambda I)^+ r_S\right] \\
&= J^\top \mathbb{E}_S\left[I_S^\top(J_S J_S^\top + \lambda I)^+ I_S\right]r \\
&= J^\top \mathbf{UDU}^\top r
\end{aligned}$$

by Lemma C.1. Then, let $D = \mathbf{D}[1:e, 1:e]$ and note also that $U = \mathbf{U}(:, 1:e)$ and $\Sigma = \mathbf{\Sigma}(1:e, 1:e)$. Then we can rewrite

$$\mathbb{E}_S\left[J_S^{+(\lambda)}r_S\right] = J^\top\mathbf{UDU}^\top r = V\Sigma U^\top\mathbf{UDU}^\top r = V\Sigma U^\top UDU^\top r = V\Sigma DU^\top r = VD\Sigma U^\top r = VDV^\top J^\top r. \tag{34}$$

The result follows since $\widetilde{W} = VDV^\top$ by Corollary C.2. □

# D. Proofs for Global Convergence of SVS-SNG

**Lemma D.1** (Variance of SNG update under SVS). *Let $J \in \mathbb{R}^{m\times n}$, $r \in \mathbb{R}^m$, and $S \sim SVS(J, k, \lambda)$. Then when $\lambda > 0$ it holds*

$$\mathbb{E}_S\left\|J_S^{+(\lambda)}r_S\right\|^2 \leq \frac{1}{\lambda}\|r\|^2. \tag{35}$$

*Proof.* Using the push-through identity $J_S^{+(\lambda)} = J_S^\top (J_S J_S^\top + \lambda I)^{-1} = (J_S^\top J_S + \lambda I)^{-1} J_S^\top$ it holds

$$\left\| J_S^{+(\lambda)} r_S \right\|^2 = r_S^\top J_S (J_S^\top J_S + \lambda I)^{-2} J_S^\top r_S$$

$$\leq \frac{1}{\lambda} r_S^\top J_S (J_S^\top J_S + \lambda I)^{-1} J_S^\top r_S$$

$$\leq \frac{1}{\lambda} \|r_S\|^2$$

Hence it follows

$$\mathbb{E}_S \left\| J_S^{+(\lambda)} r_S \right\|^2 \leq \frac{1}{\lambda} \mathbb{E}_S \|r_S\|^2 \leq \frac{1}{\lambda} \mathbb{E}_S \|r\|^2 = \frac{1}{\lambda} \|r\|^2 . \tag{36}$$

$\square$

*Proof of Theorem 4.2.* The result follows directly from (Bottou et al., 2018, Theorem 4.10) once the appropriate conditions are established. The conditions entail some continuity requirements which we have taken as assumptions (Assumption 4.1 in the reference), and some first and second moment limits which we now discuss (Assumption 4.3 in the reference).

The first moment condition follows straightforwardly from the expectation formula of Lemma 4.1 which states that

$$\mathbb{E}_S \left[ J_S^{+(\lambda)} r_S \right] = \widetilde{W} J^\top r. \tag{37}$$

Using Corollary C.2 we can rewrite this as

$$\mathbb{E}_S \left[ J_S^{+(\lambda)} r_S \right] = V D V^\top J^\top r \tag{38}$$

where $D = \mathbf{D}[1:e, 1:e]$ and $\mathbf{D}$ as in (29), and with $\lambda_{\min}(D) \geq 1/\operatorname{Tr}(JJ^\top + \lambda I)$. Since $J^\top r = \nabla_\theta \mathcal{L}(v_\theta)$ it follows that

$$\langle \nabla_\theta \mathcal{L}(v_\theta), \, \mathbb{E}_S \left[ J_S^{+(\lambda)} r_S \right] \rangle \geq \frac{1}{\operatorname{Tr}(JJ^\top + \lambda I)} \cdot \| \nabla_\theta \mathcal{L}(v_\theta) \|^2 . \tag{39}$$

Since we have assumed $v_\theta$ is a Lipschitz continuous function of $\theta$, it holds $\|J\|_2 \leq L$ for some constant $L$ and thus $\operatorname{Tr}(JJ^\top + \lambda I) \leq nL^2 + m\lambda$, and so condition (4.7a) of the reference is proven. Similarly, regarding (4.7b) we have

$$\left\| \mathbb{E}_S \left[ J_S^{+(\lambda)} r_S \right] \right\| = \left\| V D V^\top J^\top r \right\| \leq \lambda_{\max}(D) \cdot \| \nabla_\theta \mathcal{L}(v_\theta) \| , \tag{40}$$

and since $\lambda > 0$ by assumption we can upper bound

$$\lambda_{\max}(D) \leq \max_i \frac{p_{k-1}(q_{-i})}{p_k(q)} \leq \max_i \frac{p_{k-1}(q_{-i})}{q_i p_{k-1}(q_{-i}) + p_k(q_{-i})} \leq \max_i \frac{1}{q_i} \leq \frac{1}{\lambda}. \tag{41}$$

Finally, regarding the second moment condition we combine the variance bound of Lemma D.1 with the assumption on $\|r\|^2$ to obtain

$$\mathbb{E}_S \left\| J_S^{+(\lambda)} r_S \right\|^2 \leq \frac{1}{\lambda} \|r\|^2 \leq \left( \frac{C_1}{\lambda} \right) \| \nabla_\theta \mathcal{L}(v_\theta) \|^2 + \left( \frac{C_2}{\lambda} \right), \tag{42}$$

which confirms condition (4.8) of the reference. Hence all the assumptions of Theorem 4.10 of (Bottou et al., 2018) are satisfied and the theorem is proven. $\square$

**Proposition D.2.** *For any fixed instance of LLQ there exist constants $C_1, C_2$ such that $\|r\|^2 \leq C_1 \|J^\top r\|^2 + C_2$, as required by Theorem 4.2.*

*Proof.* For any matrix $X \in \mathbb{R}^{m \times m}$ let $X^{[J]}$ be the orthogonal projection of $X$ to the range of $J$, namely $UU^\top X UU^\top$ where $J = U\Sigma V^\top$ is the compact SVD of $J$. For LLQ, it holds $r = HJ\theta - b$ and thus we can bound

$$\|r\|^2 = \|HJ\theta - b\|^2 \leq 2\|HJ\theta\|^2 + 2\|b\|^2 . \tag{43}$$

The second term is already a constant and so the main task is to bound the first term. To do so note

$$\|HJ\theta\|^2 = \theta^\top J^\top H^2 J\theta = \theta^\top J^\top (H^2)^{[J]} J\theta \le C\theta^\top J^\top (H^{[J]})^2 J\theta \tag{44}$$

for some constant $C$, with the last inequality holding since $(H^2)^{[J]}$ and $(H^{[J]})^2$ both define fixed norms on $\mathrm{range}(J)$. It follows that

$$\|HJ\theta\|^2 \le C\theta^\top J^\top (H^{[J]})(H^{[J]})J\theta \le \frac{C}{\lambda^+_{\min}(JJ^\top)}\theta^\top J^\top H^{[J]}(JJ^\top)H^{[J]}J\theta = \frac{C}{\lambda^+_{\min}(JJ^\top)}\left\|J^\top HJ\theta\right\|^2. \tag{45}$$

Finally, $\left\|J^\top HJ\theta\right\|^2 = \left\|J^\top r + J^\top b\right\|^2 \le 2\left\|J^\top r\right\|^2 + 2\left\|J^\top b\right\|^2$, and as a result it suffices to choose

$$C_1 = \frac{4C}{\lambda^+_{\min}(JJ^\top)}, \quad C_2 = 2\|b\|^2 + \frac{4C}{\lambda^+_{\min}(JJ^\top)}\left\|J^\top b\right\|^2. \tag{46}$$

$\square$

# E. Proofs for Convergence Rate Analysis of SVS-SNG

**Lemma E.1** (Range containment). *If $\theta_0 \in \mathrm{range}(J^\top)$ then $\theta_t \in \mathrm{range}(J^\top)$ for all $t$.*

*Proof.* Consider the SNG iteration

$$\theta_{t+1} = \theta_t - \eta J_S^{+(\lambda)} r_S = \theta_t - \eta J^\top I_S^\top (J_S J_S^\top + \lambda I)^+ r_S. \tag{47}$$

Clearly then $\theta_t \in \mathrm{range}(J^\top) \implies \theta_{t+1} \in \mathrm{range}(J^\top)$ and the result follows by induction. $\square$

*Proof of Theorem 5.1.* For linear least-quadratics the SNG iteration takes the form

$$\theta_{t+1} = \theta_t - \eta J_S^{+(\lambda)}(H_S J\theta_t - b_S). \tag{48}$$

In the consistent case $\nabla_v \mathcal{L}(v_{\theta^*}) = 0 \implies HJ\theta^* = b$ so we can rewrite this iteration as

$$\theta_{t+1} - \theta^* = (I - \eta J_S^{+(\lambda)} H_S J)(\theta_t - \theta^*). \tag{49}$$

Now, note that since Lemma 4.1 holds for arbitrary $r$ it can be applied separately to each column of $HJ$ to show that

$$\mathbb{E}_S\left[J_S^{+(\lambda)} H_S J\right] = \widetilde{W} J^\top HJ. \tag{50}$$

Note also that although $\widetilde{W}$ is not necessarily full rank on the entirety of $\mathbb{R}^n$, Corollary C.2 shows that it *is* full rank and positive definite when restricted to $\mathrm{range}(J^\top) = \mathrm{range}(V)$, which contains all the iterates $\theta_t$ by Lemma E.1. As a result we can meaningfully measure the solution error by $\|\theta - \theta^*\|^2_{\widetilde{W}^+} = (\theta - \theta^*)^\top \widetilde{W}^+ (\theta - \theta^*)$. Using this technique to cancel out the factors of $\widetilde{W}$ in the cross-terms, we find

$$\mathbb{E}\left\|\theta_{t+1} - \theta^*\right\|^2_{\widetilde{W}^+} = (\theta_t - \theta^*)^\top \left(\widetilde{W}^+ - 2\eta J^\top HJ + \eta^2 J^\top H\, \mathbb{E}_S\left[I_S^\top (J_S^{+(\lambda)})^\top \widetilde{W}^+ J_S^{+(\lambda)} I_S\right] HJ\right)(\theta_t - \theta^*). \tag{51}$$

Using the definition of $\gamma$ it follows

$$\mathbb{E}\left\|\theta_{t+1} - \theta^*\right\|^2_{\widetilde{W}^+} \le (\theta_t - \theta^*)^\top \left(\widetilde{W}^+ - 2\eta J^\top HJ + \eta^2\gamma \cdot J^\top H^2 J\right)(\theta_t - \theta^*)$$
$$\le (\theta_t - \theta^*)^\top \left(\widetilde{W}^+ - 2\eta J^\top HJ + \eta\gamma\|H\|_2 \cdot \eta J^\top HJ\right)(\theta_t - \theta^*).$$

Hence when $\eta \le 1/(\gamma\|H\|_2)$ we obtain

$$\mathbb{E}\left\|\theta_{t+1} - \theta^*\right\|^2_{\widetilde{W}^+} \le (\theta_t - \theta^*)^\top \left(\widetilde{W}^+ - \eta J^\top HJ\right)(\theta_t - \theta^*)$$
$$\le (\theta_t - \theta^*)^\top \left(\widetilde{W}^+ - \eta \cdot \lambda_{\min}(H) \cdot J^\top J\right)(\theta_t - \theta^*).$$

Furthermore by the definition of $\widetilde{W}$ it holds

$$\widetilde{W}^+ = (J^\top J)^{1/2} \overline{P}^+ (J^\top J)^{1/2} \preceq \frac{1}{\alpha}(J^\top J), \tag{52}$$

and substituting this inequality into the above bound yields

$$\mathbb{E}\, \|\theta_{t+1} - \theta^*\|^2_{\widetilde{W}^+} \leq (\theta_t - \theta^*)^\top ([1 - \eta \cdot \lambda_{\min}(H) \cdot \alpha]\widetilde{W}^+)(\theta_t - \theta^*) = (1 - \eta \cdot \lambda_{\min}(H) \cdot \alpha)\, \|\theta_t - \theta^*\|^2_{\widetilde{W}^+}. \tag{53}$$

Finally, iterating this bound yields the theorem. $\qquad\square$

*Proof of Corollary 5.2.* We can recover the desired deterministic natural gradient descent from SNG by setting $k = m$ and $\lambda = 0$. In this case, point (3) of Proposition 5.3 shows that $\gamma \leq 1$. Additionally it holds that $\overline{P} = J^+ J$ is the orthogonal projector onto $\mathrm{range}(J^\top)$. This implies that $\alpha = \lambda^+_{\min}(\overline{P}) = 1$, and it also implies that

$$\widetilde{W} = (J^\top J)^{-1/2}\overline{P}(J^\top J)^{-1/2} = (J^\top J)^+. \tag{54}$$

Substituting these values into Theorem 5.1 yields the corollary. $\qquad\square$

*Proof of Proposition 5.3.* Consider first the case when $JJ^\top = I$. The key fact is that under this assumption it follows that $J_S J_S^\top = I$ for all $S$. This implies that $\mathrm{SVS}(J, k, \lambda)$ coincides with uniform sampling and also that $J_S^{+(\lambda)} = J_S^\top (J_S J_S^\top + \lambda I)^{-1} = \frac{1}{1+\lambda} J_S^\top$. Thus

$$\overline{P} = \mathbb{E}_S \left[ J_S^{+(\lambda)} J_S \right] = \frac{1}{1+\lambda}\mathbb{E}_S \left[ J_S^\top J_S \right] = \frac{k}{(1+\lambda)m} J^\top J, \tag{55}$$

$$\widetilde{W} = (J^\top J)^{-1/2}\overline{P}(J^\top J)^{-1/2} = \frac{k}{(1+\lambda)m}\Pi_J, \tag{56}$$

where $\Pi_J$ is the orthogonal projector onto $\mathrm{range}(J^\top)$. This implies

$$\begin{aligned}
\mathbb{E}_S \left[ I_S^\top (J_S^{+(\lambda)})^\top \widetilde{W}^+ J_S^{+(\lambda)} I_S \right] &= \frac{(1+\lambda)m}{k}\mathbb{E}_S \left[ I_S^\top (J_S^{+(\lambda)})^\top J_S^{+(\lambda)} I_S \right] \\
&= \frac{m}{(1+\lambda)k}\mathbb{E}_S \left[ I_S^\top J_S J_S^\top I_S \right] \\
&= \frac{m}{(1+\lambda)k}\mathbb{E}_S \left[ I_S^\top I_S \right] \\
&= \frac{1}{1+\lambda} I,
\end{aligned}$$

and it follows that $\gamma = 1/(1+\lambda)$.

Consider next the case when $k = 1$ and $S \sim \mathrm{SVS}(J, k, \lambda)$. In this case the sampling distribution simplifies to $p(i) = (\|J_i\|^2 + \lambda)/\mathrm{Tr}(JJ^\top + \lambda I)$, and as a result it holds

$$\overline{P} = \mathbb{E}_i \left[ J_i^{+(\lambda)} J_i \right] = \mathbb{E}_i \left[ \frac{J_i^\top J_i}{\|J_i\|^2 + \lambda} \right] = \frac{J^\top J}{\mathrm{Tr}(JJ^\top + \lambda I)}, \tag{57}$$

where here and below we can restrict $i$ to values with positive sampling probabilities to avoid zeros in the denominator. It follows that

$$\widetilde{W} = (J^\top J)^{-1/2}\overline{P}(J^\top J)^{-1/2} = \frac{1}{\mathrm{Tr}(JJ^\top + \lambda I)}\Pi_J, \tag{58}$$

where $\Pi_J$ is the orthogonal projector onto $\mathrm{range}(J^\top)$. This implies

$$
\begin{aligned}
\mathbb{E}_i \left[ I_i^\top (J_i^{+(\lambda)})^\top \widetilde{W}^+ J_i^{+(\lambda)} I_i \right] &= \mathrm{Tr}(JJ^\top + \lambda I)\, \mathbb{E}_i \left[ I_i^\top (J_i^{+(\lambda)})^\top J_i^{+(\lambda)} I_i \right] \\
&= \mathrm{Tr}(JJ^\top + \lambda I)\, \mathbb{E}_i \left[ \left( \frac{\|J_i\|^2}{(\|J_i\|^2 + \lambda)^2} \right) I_i^\top I_i \right] \\
&\preceq \mathrm{Tr}(JJ^\top + \lambda I)\, \mathbb{E}_i \left[ \frac{I_i^\top I_i}{\|J_i\|^2 + \lambda} \right] \\
&\preceq I,
\end{aligned}
$$

and it follows that $\gamma \leq 1$. As an additional comment, the inequality in the third line is tight when $\lambda = 0$, which implies that $\gamma = 1$ in that case.

Consider next the case that $k = m$, so that $J_S = J$ and $J_S^{+(\lambda)} = J^{+(\lambda)} = (J^\top J + \lambda I)^+ J^\top = J^\top (JJ^\top + \lambda I)^+$. Then

$$
\overline{P} = \mathbb{E}_S \left[ J_S^{+(\lambda)} J_S \right] = J^{+(\lambda)} J = (J^\top J + \lambda I)^+ J^\top J, \tag{59}
$$

$$
\widetilde{W} = \Pi_J (J^\top J + \lambda I)^+ \Pi_J. \tag{60}
$$

This implies

$$
\mathbb{E}_S \left[ I_S^\top (J_S^{+(\lambda)})^\top \widetilde{W}^+ J_S^{+(\lambda)} I_S \right] = (J^{+(\lambda)})^\top \widetilde{W}^+ J^{+(\lambda)} = [J(J^\top J + \lambda I)^+][\Pi_J(J^\top J + \lambda I)\Pi_J] J^{+(\lambda)} = JJ^{+(\lambda)} \leq I, \tag{61}
$$

and it follows that $\gamma \leq 1$.

Consider finally the case when $\lambda > 0$ and the sampling is uniform. As a start note that in this case, $\overline{W}$ is full rank since

$$
\overline{W} = \mathbb{E}_S \left[ I_S^\top (J_S J_S^\top + \lambda I)^{-1} I_S \right] \succeq \frac{1}{C} \mathbb{E}_S \left[ I_S^\top I_S \right] = \frac{k}{mC} I \succ 0 \tag{62}
$$

where $C = \max_S \left\| J_S J_S^\top + \lambda I \right\|_2$. Note also that by the definition of $\widetilde{W}$ it holds

$$
\widetilde{W} = (J^\top J)^{-1/2} J^\top \overline{W} J (J^\top J)^{-1/2} = V U^\top \overline{W} U V^\top \tag{63}
$$

where $U\Sigma V^\top$ is the compact SVD of $J$. Using the Rayleigh-Ritz variational principle and the fact that $V$ and $U$ have orthonormal columns it follows that

$$
\lambda_{\min}^+(\widetilde{W}) = \min_{\substack{u \in \mathrm{range}(V) \\ \|u\|=1}} u^\top V U^\top \overline{W} U V^\top u = \min_{\substack{u \in \mathbb{R}^{\mathrm{rank}(J)} \\ \|u\|=1}} u^\top U^\top \overline{W} U u \geq \min_{\substack{u \in \mathbb{R}^m \\ \|u\|=1}} u^\top \overline{W} u = \lambda_{\min}(\overline{W}), \tag{64}
$$

from which we can conclude that

$$
\left\| \widetilde{W}^+ \right\|_2 \leq \left\| \overline{W}^{-1} \right\|_2. \tag{65}
$$

We can thus calculate

$$
\begin{aligned}
\gamma &= \left\| \mathbb{E}_S \left[ I_S^\top (J_S^{+(\lambda)})^\top \widetilde{W}^+ J_S^{+(\lambda)} I_S \right] \right\|_2 \\
&\leq \left\| \overline{W}^{-1} \right\|_2 \cdot \left\| \mathbb{E}_S \left[ I_S^\top (J_S^{+(\lambda)})^\top J_S^{+(\lambda)} I_S \right] \right\|_2 \\
&= \left\| \overline{W}^{-1} \right\|_2 \cdot \left\| \mathbb{E}_S \left[ I_S^\top (J_S J_S^\top + \lambda I)^+ J_S J_S^\top (J_S J_S^\top + \lambda I)^+ I_S \right] \right\|_2 \\
&\leq \left\| \overline{W}^{-1} \right\|_2 \cdot \left\| \mathbb{E}_S \left[ I_S^\top (J_S J_S^\top + \lambda I)^+ I_S \right] \right\|_2 \\
&= \left\| \overline{W}^{-1} \right\|_2 \cdot \left\| \overline{W} \right\|_2 \\
&= \kappa(\overline{W}).
\end{aligned}
$$

Invoking Theorem 5.1 of (Goldshlager et al., 2025) then yields the desired bound. $\qquad\square$

## F. Convergence Rate Under Strong Compatibility

In this appendix we provide an additional proposition which shows that the influence of $\gamma$ can disappear entirely when $J$ and $H$ satisfy a strong compatibility condition. The result is that in at least one special case, the advantages of sketch-and-project methods can extend to a genuine natural gradient setting without qualification.

**Proposition F.1** (Convergence under strong compatibility). *Consider a consistent instance of linear least-quadratics and assume an initial guess $\theta_0 \in \operatorname{range}(J^\top)$. Set the sampling distribution to be SVS($J, k, \lambda$) and **assume a strong compatibility condition** $[H, JJ^\top] = 0$ where $[X, Y] = XY - YX$ is the commutator. Then with step size $\eta \leq 1/\|H\|_2$ the SNG iterates satisfy*

$$\mathbb{E} \|\theta_t - \theta^*\|^2 \leq (1 - \eta \cdot \lambda_{\min}(H) \cdot \alpha)^t \|\theta_0 - \theta^*\|^2, \tag{66}$$

*with $\alpha$ as in (13).*

We now prove two auxiliary lemmas before proceeding to prove the proposition.

**Lemma F.2** (Projected Hessian). *Let $H \succ 0$, assume $[H, JJ^\top] = 0$, and define $\widetilde{H} = J^+ H J$. It follows that*

$$\widetilde{H} = V \Lambda V^\top \tag{67}$$

*where $U\Sigma V^\top$ is the compact SVD of $J$ and $\Lambda$ is diagonal with $\lambda_{\min}(H) \leq \lambda_{\min}(\Lambda) \leq \lambda_{\max}(\Lambda) \leq \lambda_{\max}(H)$.*

*Proof.* By the assumptions $H \succ 0$ and $[H, JJ^\top] = 0$, $H$ admits an orthogonal decomposition $H = U\Lambda U^\top + H_\perp$ where $\Lambda \succ 0$ is diagonal, $H_\perp \succeq 0$ and $J^\top H_\perp = H_\perp J = 0$. It follows that

$$\widetilde{H} = J^+ H J = V \Lambda V^\top. \tag{68}$$

Furthermore, the eigenvalues of $\Lambda$ are a subset of the eigenvalues of $H$ due to the orthogonal decomposition and are thus contained within $[\lambda_{\min}(H), \lambda_{\max}(H)]$. $\square$

**Lemma F.3** (Expected projection). *Let $S \sim$ SVS($J, k, \lambda$) and define $\alpha = \lambda_{\min}^+(\overline{P})$, $\overline{P} = \mathbb{E}_S[P(S)]$, $P(S) = J_S^{+(\lambda)} J_S$. It follows that*

$$\overline{P} = V \Lambda V^\top \tag{69}$$

*where $U\Sigma V^\top$ is the compact SVD of $J$ and $\Lambda \succeq \alpha I$ is diagonal.*

*Proof.* Using Lemma C.1 and defining $D = \mathbf{D}[1 : e, 1 : e]$ we have

$$\begin{aligned}
\overline{P} &= \mathbb{E}_S \left[ J_S^{+(\lambda)} J_S \right] \\
&= J^\top \mathbb{E}_S \left[ I_S^\top (J_S J_S^\top + \lambda I)^+ I_S \right] J \\
&= J^\top \mathbf{U}\mathbf{D}\mathbf{U}^\top J \\
&= V(\Sigma U^\top \mathbf{U}\mathbf{D}\mathbf{U}^\top U\Sigma)V^\top \\
&= V(\Sigma D\Sigma)V^\top.
\end{aligned}$$

The result follows by defining $\Lambda = \Sigma D\Sigma$. $\square$

*Proof of Proposition F.1.* To start define $\widetilde{H} = J^+ H J$ and let $U\Sigma V^\top$ be the compact SVD of $J$. For linear least-quadratics the SNG iteration takes the form

$$\theta_{t+1} = \theta_t - \eta J_S^{+(\lambda)}(H_S J\theta_t - b_S). \tag{70}$$

In the consistent case $\nabla_v \mathcal{L}(v_{\theta^*}) = 0 \implies H J\theta^* = b$ so we can rewrite this iteration as

$$\theta_{t+1} - \theta^* = (I - \eta J_S^{+(\lambda)} H_S J)(\theta_t - \theta^*). \tag{71}$$

Using the assumption $[H, JJ^\top] = 0$ it holds $\operatorname{range}(HJ) = \operatorname{range}(J)$ and thus

$$H_S J = I_S H J = I_S J J^+ H J = J_S \widetilde{H}. \tag{72}$$

This leads to the equivalent form

$$\theta_{t+1} - \theta^* = (I - \eta J_S^{+(\lambda)} J_S \widetilde{H})(\theta_t - \theta^*) = (I - \eta P(S)\widetilde{H})(\theta_t - \theta^*). \tag{73}$$

Taking squared norms, adding weights and expectations we obtain

$$\mathbb{E}\,\|\theta_{t+1} - \theta^*\|^2 = (\theta_t - \theta^*)^\top \left( I - 2\eta \overline{P}\widetilde{H} + \eta^2 \widetilde{H}\mathbb{E}_S\left[P(S)^2\right]\widetilde{H} \right)(\theta_t - \theta^*)$$

$$\leq (\theta_t - \theta^*)^\top \left( I - 2\eta \overline{P}\widetilde{H} + \eta^2 \widetilde{H}\overline{P}\widetilde{H} \right)(\theta_t - \theta^*)$$

using $P(S) \preceq I \implies P(S)^2 \preceq P(S)$. Now, using Lemmas F.2 and F.3 we see that $\overline{P}$ commutes with $\widetilde{H}$ and thus when $\eta \leq 1/\|\widetilde{H}\|_2$ it holds $\eta^2 \widetilde{H}\overline{P}\widetilde{H} \preceq \eta\overline{P}\widetilde{H}$, leading to the simpler bound

$$\mathbb{E}\,\|\theta_{t+1} - \theta^*\|^2 \leq (\theta_t - \theta^*)^\top \left( I - \eta \overline{P}\widetilde{H} \right)(\theta_t - \theta^*). \tag{74}$$

Again applying Lemmas F.2 and F.3 to write $\overline{P} = V\Lambda_P V^\top$, $\widetilde{H} = V\Lambda_H V^\top$ we have

$$\mathbb{E}\,\|\theta_{t+1} - \theta^*\|^2 \leq (\theta_t - \theta^*)^\top \left( I - \eta V\Lambda_P\Lambda_H V^\top \right)(\theta_t - \theta^*). \tag{75}$$

By Lemma E.1 it holds $\theta_t - \theta^* \in \mathrm{range}(V)$ and thus

$$\mathbb{E}\,\|\theta_{t+1} - \theta^*\|^2 \leq (1 - \eta \cdot \lambda_{\min}(\Lambda_P) \cdot \lambda_{\min}(\Lambda_H))\,\|\theta_t - \theta^*\|^2. \tag{76}$$

Using $\lambda_{\min}(\Lambda_P) \geq \alpha$, $\lambda_{\min}(\Lambda_H) \geq \lambda_{\min}(H)$ it follows

$$\mathbb{E}\,\|\theta_{t+1} - \theta^*\|^2 \leq (1 - \eta \cdot \lambda_{\min}(H) \cdot \alpha)\,\|\theta_t - \theta^*\|^2, \tag{77}$$

and iterating this bound yields the proposition. $\qquad\square$

## G. Convergence Rate Under Smoothness, Strong Convexity, and Inconsistency

In this appendix we provide an extension of Theorem 5.1 to the more general case when $\mathcal{L}$ is smooth and strongly convex and the problem is possibly inconsistent. The results are qualitatively similar to the consistent quadratic case.

**Theorem G.1** (Extension of Theorem 5.1). *Consider a parametric optimization problem $\min_\theta \mathcal{L}(v_\theta)$. Assume that $\mathcal{H} = \mathbb{R}^m$ and that $v_\theta = J\theta$ is linear while $\mathcal{L}$ is $L$-smooth and $\sigma$-strongly convex as a function of $v$. Assume an initial guess of $\theta_0 \in \mathrm{range}(J^\top)$, let $\theta^*$ be the unique minimizer within $\mathrm{range}(J^\top)$, and define $\alpha, \gamma$ as in Theorem 5.1.*

*Then when $S \sim SVS(J, k, \lambda)$ and $0 < \eta \leq \frac{1}{2\gamma L}$, it holds*

$$\mathbb{E}\,\|\theta_t - \theta^*\|^2_{\widetilde{W}^+} \leq (1 - \eta \cdot \sigma \cdot \alpha)^t\,\|\theta_0 - \theta^*\|^2_{\widetilde{W}^+} + \frac{2\gamma\eta}{\sigma\alpha}\,\|r^*\|^2 \tag{78}$$

*with $r^* = r(\theta^*)$.*

The result of Theorem G.1 is that even in this more general setting, the convergence rate of SNG is still governed by the conditioning of $\mathcal{L}$ and the sketch-and-project structure induced by $J$. It is notable that in the consistent case, it holds $r^* = 0$ and so a linear convergence rate is recovered. The proof follows similarly to the proof of Theorem 5.1 but also incorporates ideas from the proof of Theorem 5.8 of (Garrigos & Gower, 2023), with the difference being that we take care to apply the smoothness and strong convexity properties to only the function space loss $\mathcal{L}$ rather than the entire loss $\mathcal{L}(v_\theta)$.

*Proof.* Recall the SNG update

$$\theta_{t+1} = \theta_t - \eta J_S^{+(\lambda)} r_S, \tag{79}$$

where $r = \nabla_v \mathcal{L}(v_{\theta_t})$. It follows that, conditioning on $\theta_t$,

$$\mathbb{E}\left[\|\theta_{t+1} - \theta^*\|^2_{\widetilde{W}^+} \mid \theta_t\right] = \mathbb{E}_S \left\|\theta_t - \eta J_S^{+(\lambda)} r_S - \theta^*\right\|^2_{\widetilde{W}^+} \tag{80}$$

$$= \|\theta_t - \theta^*\|^2_{\widetilde{W}^+} - 2\eta(\theta_t - \theta^*)^\top \widetilde{W}^+ \mathbb{E}_S\left[J_S^{+(\lambda)} r_S\right] + \eta^2 \mathbb{E}_S \left\|J_S^{+(\lambda)} r_S\right\|^2_{\widetilde{W}^+} \tag{81}$$

$$\leq \|\theta_t - \theta^*\|^2_{\widetilde{W}^+} - 2\eta(\theta_t - \theta^*)^\top J^\top r + \gamma\eta^2\,\|r\|^2, \tag{82}$$

where the last step uses Lemma 4.1, the definition of $\gamma$, and the fact that $\widetilde{W}^+\widetilde{W}J^\top = J^\top$ due to Corollary C.2.

Now using the strong convexity of $\mathcal{L}$ we can bound

$$\mathcal{L}(v_{\theta^*}) - \mathcal{L}(v_{\theta_t}) \geq (v_{\theta^*} - v_{\theta_t})^\top r + \frac{\sigma}{2}\|v_{\theta^*} - v_{\theta_t}\|^2 = (\theta^* - \theta_t)^\top J^\top r + \frac{\sigma}{2}\|\theta_t - \theta^*\|^2_{J^\top J}. \tag{83}$$

Rearranging and using (52) yields

$$(\theta_t - \theta^*)^\top J^\top r \geq \mathcal{L}(v_{\theta_t}) - \mathcal{L}(v_{\theta^*}) + \frac{\sigma\alpha}{2}\|\theta_t - \theta^*\|^2_{\widetilde{W}^+}. \tag{84}$$

Additionally it holds

$$\|r\|^2 \leq 2\|r - r^*\|^2 + 2\|r^*\|^2 = 2\|\mathcal{L}'(v_{\theta_t}) - \mathcal{L}'(v_{\theta^*})\|^2 + 2\|r^*\|^2. \tag{85}$$

Invoking Lemma 2.29 of (Garrigos & Gower, 2023) on the first term yields

$$\begin{aligned}
2\|\mathcal{L}'(v_{\theta_t}) - \mathcal{L}'(v_{\theta^*})\|^2 &\leq 4L\left[\mathcal{L}(v_{\theta_t}) - \mathcal{L}(v_{\theta^*}) - \mathcal{L}'(v_{\theta^*})^\top(v_{\theta_t} - v_{\theta^*})\right] \\
&= 4L\left[\mathcal{L}(v_{\theta_t}) - \mathcal{L}(v_{\theta^*}) - (r^*)^\top J(\theta_t - \theta^*)\right] \\
&= 4L\left[\mathcal{L}(v_{\theta_t}) - \mathcal{L}(v_{\theta^*})\right]
\end{aligned}$$

using the optimality condition $J^\top r^* = \nabla_\theta \mathcal{L}(v_{\theta^*}) = 0$. It follows that

$$\|r\|^2 \leq 4L\left(\mathcal{L}(v_{\theta_t}) - \mathcal{L}(v_{\theta^*})\right) + 2\|r^*\|^2. \tag{86}$$

Substituting (86) and (84) into the error bound (82) yields the conditional expectation bound

$$\mathbb{E}\left[\|\theta_{t+1} - \theta^*\|^2_{\widetilde{W}^+} \mid \theta_t\right] \leq (1 - \eta \cdot \sigma \cdot \alpha)\|\theta_t - \theta^*\|^2_{\widetilde{W}^+} + 2\eta(2\gamma\eta L - 1)(\mathcal{L}(v_{\theta_t}) - \mathcal{L}(v_{\theta^*})) + 2\gamma\eta^2\|r^*\|^2. \tag{87}$$

As long as $\eta \leq \frac{1}{2\gamma L}$, the second term is nonpositive and the bound simplifies to

$$\mathbb{E}\left[\|\theta_{t+1} - \theta^*\|^2_{\widetilde{W}^+} \mid \theta_t\right] \leq (1 - \eta \cdot \sigma \cdot \alpha)\|\theta_t - \theta^*\|^2_{\widetilde{W}^+} + 2\gamma\eta^2\|r^*\|^2. \tag{88}$$

Given that (88) holds for any admissible current iterate $\theta_t \in \text{range}(J^\top)$, it must hold $\eta\sigma\alpha \leq 1$, since otherwise it would be possible to take $\theta_t - \theta^*$ large enough to make the right-hand side of (88) negative, yielding a contradiction. With this guarantee we can safely take total expectations and iterate (88) to obtain

$$\mathbb{E}\|\theta_t - \theta^*\|^2_{\widetilde{W}^+} \leq (1 - \eta \cdot \sigma \cdot \alpha)^t\|\theta_0 - \theta^*\|^2_{\widetilde{W}^+} + 2\gamma\eta^2\|r^*\|^2 \sum_{j=0}^{t-1}(1 - \eta\sigma\alpha)^j. \tag{89}$$

Using again the fact that $\eta\sigma\alpha \leq 1$, we can upper bound the geometric series by its infinite sum $(\eta\sigma\alpha)^{-1}$ to obtain the final result:

$$\mathbb{E}\|\theta_t - \theta^*\|^2_{\widetilde{W}^+} \leq (1 - \eta \cdot \sigma \cdot \alpha)^t\|\theta_0 - \theta^*\|^2_{\widetilde{W}^+} + \frac{2\gamma\eta}{\sigma\alpha}\|r^*\|^2. \tag{90}$$

$\square$

## H. Proof for SPRING as an Accelerated Sketch-and-Project Method

*Proof of Theorem 6.1.* The recent work of (Dereziński et al., 2025b) was the first to incorporate regularization into an accelerated sketch-and-project algorithm. That work focused on the case of randomized block Kaczmarz methods but their formulation extends trivially to general sketch-and-project algorithms. In this more general context their method takes the form

$$\begin{aligned}
w_t &= (\Omega_t J)^{+(\lambda)}(\Omega_t J\widetilde{\theta}_t - \Omega_t b), \\
\widetilde{\phi}_{t+1} &= \mu(\widetilde{\phi}_t - w_t), \\
\widetilde{\theta}_{t+1} &= \widetilde{\theta}_t - w_t + \widetilde{\eta}\widetilde{\phi}_{t+1}.
\end{aligned} \tag{91}$$

We use the symbols $\widetilde{\theta}, \widetilde{\phi}, \widetilde{\eta}$ since these variables do not correspond directly to the variables $\theta, \phi, \eta$ of our formulation. We also recall for convenience the formulation that we have proposed, recalling from equation (20) the following iteration:

$$
\begin{aligned}
\phi_{t+1} &= \mu\phi_t + (\Omega_t J)^{+(\lambda)}(\Omega_t b - \Omega_t J(\theta_t + \mu\phi_t)), \\
\theta_{t+1} &= \theta_t + \eta\phi_{t+1}.
\end{aligned}
\tag{92}
$$

We next show that the algorithm (91) with initial guess $\widetilde{\theta}_0 = \theta_0$, initial momentum $\widetilde{\phi}_0 = 0$, momentum coefficient $\mu$, and step size $\widetilde{\eta}$ is equivalent to the algorithm (92) with initial guess $\theta_0$, initial momentum $\phi_0 = 0$, momentum coefficient $\mu$, and step size $\eta = 1 + \mu(\widetilde{\eta} - 1)$ in that the resulting iterates satisfy $\widetilde{\phi}_t = \mu\phi_t$, $\widetilde{\theta}_t = \theta_t + \mu\phi_t$.

Proceeding by induction, as a base case we have $\widetilde{\phi}_0 = \mu\phi_0 = 0$ and $\widetilde{\theta}_0 = \theta_0 + \mu\phi_0 = \theta_0$. Now, suppose these identities hold for some fixed $t \geq 0$. Then calculate using (91) and (92)

$$
\begin{aligned}
\widetilde{\phi}_{t+1} &= \mu(\widetilde{\phi}_t - (\Omega_t J)^{+(\lambda)}(\Omega_t J\widetilde{\theta}_t - \Omega_t b)) \\
&= \mu(\mu\phi_t - (\Omega_t J)^{+(\lambda)}(\Omega_t J(\theta_t + \mu\phi_t) - \Omega_t b)) \\
&= \mu(\mu\phi_t + (\Omega_t J)^{+(\lambda)}(\Omega_t b - \Omega_t J(\theta_t + \mu\phi_t))) \\
&= \mu\phi_{t+1}.
\end{aligned}
$$

Using the above, proceed to verify

$$
\begin{aligned}
\widetilde{\theta}_{t+1} &= \widetilde{\theta}_t - (\Omega_t J)^{+(\lambda)}(\Omega_t J\widetilde{\theta}_t - \Omega_t b) + \widetilde{\eta}\widetilde{\phi}_{t+1} \\
&= \theta_t + \mu\phi_t - (\Omega_t J)^{+(\lambda)}(\Omega_t J(\theta_t + \mu\phi_t) - \Omega_t b) + \mu\widetilde{\eta}\phi_{t+1} \\
&= \theta_t + (1 + \mu\widetilde{\eta})\phi_{t+1} \\
&= (\theta_t + \eta\phi_{t+1}) + \mu\phi_{t+1} \\
&= \theta_{t+1} + \mu\phi_{t+1}.
\end{aligned}
$$

This completes the inductive step.

We now turn our attention to the SPRING algorithm. In particular we consider applying the new accelerated sketch-and-project step of equation (92) to the natural gradient subproblem (3) with the row sampling sketching matrix $I_S$. We should be careful about the step sizes here since as defined, both the natural gradient subproblem *and* the accelerated sketch-and-project step have step size parameters. As such let us use the notation $\eta_1$ to denote the step size of the subproblem and $\eta_2$ to denote the step size of the accelerated sketch-and-project step. We then obtain the update formulas

$$
\begin{aligned}
\phi_{t+1} &= \mu\phi_t + J_S^{+(\lambda)}((-\eta_1 r + J\theta_t)_S - J_S(\theta_t + \mu\phi_t)) \\
&= \mu\phi_t + J_S^{+(\lambda)}(-\eta_1 r_S - \mu J_S\phi_t), \\
\theta_{t+1} &= \theta_t + \eta_2\phi_{t+1}.
\end{aligned}
\tag{93}
$$

Rescaling $\phi \leftarrow -\phi/\eta_1$ we can rewrite this as

$$
\begin{aligned}
\phi_{t+1} &= \mu\phi_t + J_S^{+(\lambda)}(r_S - \mu J_S\phi_t), \\
\theta_{t+1} &= \theta_t - \eta_1\eta_2\phi_{t+1}.
\end{aligned}
\tag{94}
$$

Finally, joining the step sizes using $\eta = \eta_1\eta_2$ yields the standard form of SPRING:

$$
\begin{aligned}
\phi_{t+1} &= \mu\phi_t + J_S^{+(\lambda)}(r_S - \mu J_S\phi_t), \\
\theta_{t+1} &= \theta_t - \eta\phi_{t+1}.
\end{aligned}
\tag{95}
$$

$\square$

## I. Additional Experiments Regarding Squared Volume Sampling as a Proxy

In this appendix we provide two additional experiments that confirm the suitability of SVS-SNG as a theoretical proxy for realistic SNG algorithms. First, in Figure 6, we test the same three algorithms as in Figure 2 for a simple instance of

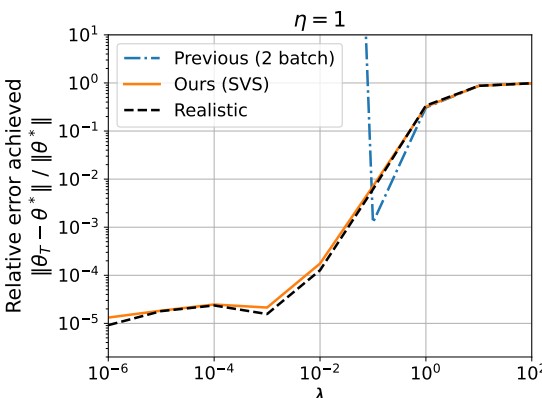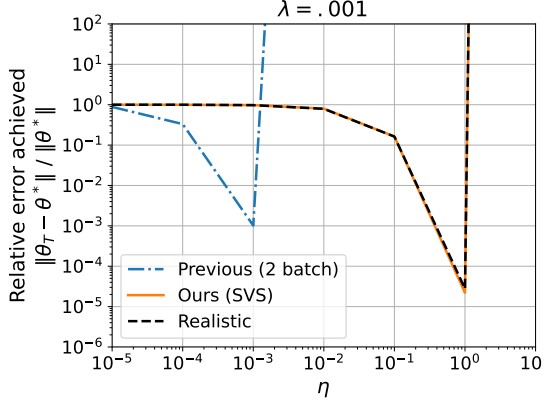

*Figure 6.* Empirical performance of two theoretical proxies for SNG relative to a realistic algorithm using a single, uniformly sampled mini-batch. The results are for a consistent instance of linear least-quadratics involving random Gaussian matrices and the behavior is tested for a fixed step size $\eta$ and varying regularizations $\lambda$ (left), as well as for fixed $\lambda$ and varying $\eta$ (right). In both cases the behavior of SVS-SNG matches closely with the realistic algorithm, while the behavior of the two mini-batch proxy is entirely distinct. The dimensions are $m = 10^3$, $n = 10^2$, and $k = 10$, which is small enough to directly implement SVS. The $y$-axis represents the relative error achieved after $T = 10^3$ iterations.

linear least-quadratics involving random Gaussian data matrices. Second, in Figure 7, we re-run the same experiments as in Figure 6 but with a single row of the Jacobian inflated by a factor of 100 in order to break the Gaussian structure. The results in both cases show that SVS-SNG is a much better proxy for the realistic SNG algorithm than the previously studied algorithm with two independent mini-batches, just like in Figure 2.

Interestingly, the only case when the behavior of the proxy differs substantially from the behavior of the realistic algorithm is when $\lambda$ is taken to be quite small in the presence of the row inflation. In this case the proxy performs better than the realistic algorithm, which highlights the potential benefits of SVS-inspired sampling schemes. In contrast, when the row inflation is present the two mini-batch algorithm becomes unable to make progress in reducing the solution error under any of the tested hyperparameter settings.

## J. Details of Numerical Experiments

In this appendix we provide the details for our numerical experiments. The code to reproduce all the results can be found at https://github.com/ggoldsh/sketch-and-project-natural-gradient.

### J.1. Figure 2

For Figure 2 the goal is to solve a discrete version of the Poisson problem

$$-\Delta v = b \tag{96}$$

on the domain $\Omega = [0, 2\pi]$ with periodic boundary conditions. We use $m = 100$ grid points $x_1, \ldots, x_m \in \Omega$, which leads to a grid spacing of $h = 2\pi/m$ and a discrete Laplacian

$$-\Delta \approx H = \frac{1}{h^2} \begin{bmatrix} 2 & -1 & \cdots & -1 \\ -1 & 2 & \cdots & 0 \\ \vdots & \vdots & \ddots & \vdots \\ 0 & \cdots & 2 & -1 \\ -1 & \cdots & -1 & 2 \end{bmatrix}. \tag{97}$$

We define the discrete solution as

$$v^* = \begin{bmatrix} f(x_1) \\ \vdots \\ f(x_m) \end{bmatrix} ; \quad f(x) = \sin(2x) + 2\cos(3x) + \sin(5x), \tag{98}$$

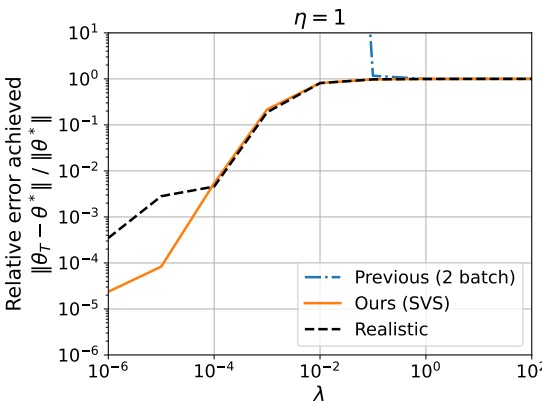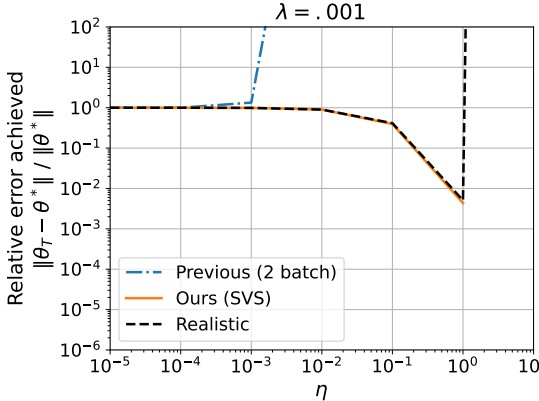

*Figure 7.* Empirical performance of two theoretical models for SNG relative to a realistic algorithm using a single, uniformly sampled mini-batch. The results are for consistent instances of linear least-quadratics, the Jacobian $J$ is a random Gaussian matrix with its first row inflated by a factor of 100, and the behavior is tested for fixed $\eta$ and varying $\lambda$ (left) as well as fixed $\lambda$ and varying $\eta$ (right). In both cases the behavior of SVS-SNG matches qualitatively with the behavior of the realistic algorithm, while the behavior of the previously studied algorithm with two independent mini-batches is entirely distinct. The dimensions are $m = 10^3$, $n = 10^2$, and $k = 10$, which is small enough to directly implement SVS. The $y$-axis represents the relative error achieved after $T = 10^3$ iterations.

and the discrete right-hand side via $b = Hv^*$.

Using these quantities we define the loss function

$$\mathcal{L}(v) = \frac{1}{2}v^\top Hv - v^\top b \qquad (99)$$

which by construction has $v^*$ as an exact minimizer. We then try to recover $v^*$ by applying various subsampled natural gradient algorithms to the parametric optimization problem $\min_\theta \mathcal{L}(v_\theta)$ where $v_\theta$ is represented by a small neural network. In fact the problem (99) has a one-dimensional family of minimizers since the discrete Laplacian annihilates all constant functions. To account for this degeneracy we always shift both $v^*$ and $v_\theta$ so that their entries sum to zero before computing $L_2$ errors.

The neural network is a simple periodic ResNet (He et al., 2016) with 5 layers, which takes as input an index $1 \le i \le m$ and outputs the corresponding entry $v_\theta[i]$. In particular given an input $i$ the output is defined by

$$y_1 = \tanh\left(W_1 \begin{bmatrix} \cos(x_i) \\ \sin(x_i) \end{bmatrix} + b_1 \right), \qquad (100)$$

$$y_i = y_{i-1} + \tanh(W_i y_{i-1} + b_i); \ i = 2, 3, 4, \qquad (101)$$

$$v_\theta[i] = W_5 y_4 + b_5. \qquad (102)$$

The intermediate layers have 50 neurons each and the weight matrices $W_i$ and bias vectors $b_i$ have the appropriate dimensions to match. The weights are initialized using a LeCun normal initialization and the biases are all initialized to zero. The parameters $\{W_i, b_i\}_{i=1}^5$ are collected into a single vector $\theta \in \mathbb{R}^n$ with $n = 7801$. During the optimization, the subsampled Jacobian $J_S = \nabla_\theta[(v_\theta)_S]$ is calculated via automatic differentiation, while the subsampled function space gradient is calculated using only forward evaluations of $v_\theta$ at the indices in $S$ as well as their neighbors, which suffices for reconstructing $r_S = (Hv_\theta - b)_S$ due to the local structure of $H$.

### J.2. Figures 6 and 7

For both Figures 6 and 7 $J$ is initially constructed with independent standard normal entries. For Figure 7 only, the first row of $J$ is then multiplied by 100 to introduce some hidden structure which can be adversarial for the case of uniform sampling. Finally, in both cases the matrix $J$ is normalized to have a maximum singular value of 1.

For both Figures 6 and 7 the $H$ matrix is constructed as

$$H = LL^\top / \left\| LL^\top \right\|_2 \qquad (103)$$

where $L$ is an $m \times m$ matrix with standard random normal entries. The solution $\theta^*$ is then also constructed with standard random normal entries and the linear term of the loss is constructed as $b = HJ\theta^*$ to ensure consistency.

### J.3. Figure 3

For Figure 3, $J$ is the only matrix that must be instantiated, and it is constructed as

$$J = Q_1 D Q_2 \tag{104}$$

where $Q_1$ is a random $m \times n$ matrix with orthonormal columns, $D$ is a diagonal matrix with $D_{ii} = 1/i^2$, and $Q_2$ is a random $n \times n$ orthogonal matrix. Given $J$ and assuming SVS, we can compute $\overline{P}$ directly using elementary symmetric polynomials; see for example the proof of Lemma F.3. This enables a direct computation of $\alpha$ as well. For $\gamma$, we draw samples from $\text{SVS}(J, k, \lambda)$ to estimate the required matrix

$$X = \mathbb{E}_S \left[ I_S^\top (J_S^{+(\lambda)})^\top \widetilde{W}^+ J_S^{+(\lambda)} I_S \right], \tag{105}$$

and we then directly compute the norm of $X$. To provide a reasonable estimation of $X$ across all sample sizes, we allocated an equal time budget of five minutes for each value of $k$, which resulted in sample counts $[23503, 24946, 20586, 21017, 6610]$, respectively. For $k = 1$ we already know that $\gamma = 1$ by Proposition 5.3 so we simply plot this value without any sampling.

### J.4. Figure 4

For Figure 4 the Jacobian is constructed as

$$J = Q_1 D Q_2 \tag{106}$$

where $Q_1$ is a random $m \times n$ matrix with orthonormal columns, $D$ is a diagonal matrix with $D_{ii} = 1/i^2$, and $Q_2$ is a random $n \times n$ orthogonal matrix. For the inset the set-up is the same except that $D = I$. The quantities $H, \theta^*, b$ are constructed exactly as for Figures 6 and 7 (see for example (103)). To estimate the convergence rate for each value of $k$ we fit a linear function to the logarithm of the solution errors ($\|\theta_0 - \theta^*\|^2, \|\theta_1 - \theta^*\|^2, \ldots \|\theta_T - \theta^*\|^2$) and use the slope of the fit as an estimate for the empirical convergence rate.

### J.5. Figure 5

For Figure 5 we construct

$$J = Q_1 D Q_2 \tag{107}$$

where $Q_1$ is a random $m \times n$ matrix with orthonormal columns, $D$ is a diagonal matrix with $D_{ii} = 1/i$, and $Q_2$ is a random $n \times n$ orthogonal matrix. We further construct

$$H = Q D Q^\top \tag{108}$$

where $Q$ is a random $m \times m$ orthogonal matrix and $D$ is a diagonal $m \times m$ matrix with

$$D_{ii} = \frac{1}{1 + (i - 1) \cdot \frac{n-1}{m-1}} \tag{109}$$

so that $D_{11} = 1$, $D_{mm} = 1/n$, and $\kappa(H) = n = 100$. From there the values of $\theta^*$ and $b$ are constructed as in the other experiments ($\theta$ random normal, $b$ for consistency).

To tune $\eta$ for SNG we use a dynamic grid search starting from $\eta = 1$ and exploring values on a logarithmic grid ($\eta = 2^i$ for various integers $i$) until we find a value that performs better than any other value within 2 grid steps. Similarly for SPRING we tune $\eta$ and $\mu$ by starting at $\eta = 1$, $\mu = 0$ and exploring values on a logarithmic grid ($\eta = 2^i$, $\mu = 1 - 2^{-j}$ for various integers $i$ and nonnegative integers $j$) until we find a value that performs better than any other value within at most 2 grid steps in each dimension. Once the optimal hyperparameters are found, the empirical rate constants are measured in the same way as for Figure 4.

## Disclaimer

