# OpenReview forum: "A Sketch-and-Project Analysis of Subsampled Natural Gradient Algorithms"
_ICML.cc/2026/Conference — ICML 2026 regular_

### Official Review · Reviewer_G6kF · 2026-03-08

**Soundness:** 3
**Presentation:** 4
**Significance:** 3
**Originality:** 1
**Overall Recommendation:** 3
**Confidence:** 3

**Summary:**

This paper presents a different way of analyzing what looks like subsampled Gauss Newton / block kaczmarz method, which they term subsampled natural gradient descent. Unsurprisingly, it does better than SGD when the spectrum of the Jacobian is not that uniform. It makes the case that a direct analysis of this  method is challenging, because the stochasticity of the gradient and preconditioner is coupled. So, through some techniques,  this paper produces a different (but equivalent) formulation which looks more like a stochastic quadratic minimization, but where the quadratic matrix contains all the preconditioning terms. Specifically, the sampling itself is via a distribution SVS, which puts more weight on sets which contribute more to the preconditioner, in a volume sense. (This looks a lot like DPP.) Then one can put conditions on this matrix, like spectral requirements, and guarantee strong convergence rates.

**Compliance With Llm Reviewing Policy:**

Affirmed.

**Final Justification:**

My concerns were not mitigated so I keep my score

**Key Questions For Authors:**

Can you discuss a bit the computational complexity of this SVS? Is it combinatorial? Is there a difference between uniform sampling with or without replacement in practice? In theory?

**Limitations:**

yes

**Strengths And Weaknesses:**

I think indeed a lot of the magic is happening with the spectral bounds on the subsampled matrices under SVS. I understand they are mostly from preexisting work, but it is a bit unnerving to see almost all proofs of these bounds diverted to another paper, which also seem recent. Maybe the paper or appendix can introduce some proof summaries or intuitions, so the results do not appear too magical, e.g. A.1, and lower bound of $\lambda_\min(D)$ in Lemma A.2. It does indeed feel that once the spectrum of $\tilde W$ is well characterized, the rest is just standard rates -- especially so since it's restricted to quadratic problems.  (If there are more nontrivial challenges, please also explain, maybe in the paper intro.)

Similarly, Thm. 6.1 seems more about showing the equivalence of a method to SPRING than computing accelerated rates for a different method.

Overall, I have no qualms about soundness, and actually  I think the main paper is written very well and is very easy to follow, and even the significance is interesting enough that it should be considered. But I think the main weakness of this paper is originality.

---

> ### Author Rebuttal · Authors · 2026-03-30
>
> We thank the reviewer for their time and attention. We clarify that subsampled natural gradient algorithms are distinct from both subsampled Gauss-Newton and randomized block Kaczmarz methods. For general parametric optimization problems the Gauss-Newton preconditioner is $J^\top H J$ where $H=\nabla^2_v \mathcal{L}(v_\theta)$ is the function space Hessian, whereas the natural gradient preconditioner is $J^\top J$. Furthermore, block Kaczmarz methods apply only to linear systems and least-squares problems, whereas natural gradient methods apply  to all types of parametric optimization problems. Finally, it is important to note that our SNG method is not arbitrary or contrived but rather an important practical algorithm that is widely adopted in one subfield of machine learning (neural wavefunctions) and promising in another (physics-informed neural networks), yet has thus far lacked any theoretical explanation. This provides the context and motivation for our contributions.
>
> **Originality:**
>
> We respectfully disagree with the concern about the lack of originality. While technically our proofs build on existing machinery, our original contributions include:
> 1. Showing that it is possible to analyze SNG without decoupling the gradient and the preconditioner.
> 2. Identifying squared volume sampling as a suitable assumption to facilitate the analysis.
> 3. Providing the first characterization of the convergence rate of SNG in the small-sample setting.
> 4. Providing the first principled derivation of the SPRING algorithm, leading to the first convergence guarantee for SPRING.
>
> We note that these contributions are aligned with the definition of originality in the ICML reviewer guidelines, which allows for works that introduce novel combinations of existing techniques or that provide new insights or introduce new perspectives on existing methods. To further highlight the conceptual novelty, we emphasize that it is widely believed that running SNG with a single mini-batch is merely a practical hack, while the principled approach is to use two independent mini-batches. Relative to this belief, our work makes a significant departure by arguing that the two mini-batch approach is fundamentally flawed and that a better theoretical idealization is provided by preserving the sketch-and-project structure and adopting squared volume sampling.
>
> **Other comments and questions**:
> 1. Too much magic: we thank you for the feedback. At a high level these results are proved by applying Cramer's rule and then carefully calculating sums of either determinants or adjoints of principal submatrices, which turn out to have nice formulas in terms of symmetric polynomials of the eigenvalues. In the final version we will provide more details and intuition.
> 2. Complexity of SVS: in fact the complexity is much better than combinatorial. The most naive approach relies on an eigenvalue decomposition of $JJ^\top + \lambda I$ with complexity  $O(m^3)$. There are also MCMC approaches with complexity $O(m\cdot {\rm poly}(k))$ [AGR16], and oversample-and-subselect approaches which can be even faster under favorable conditions  [CDV20]. See [CDV20], page 2 for a summary of these and other approaches. Thus, while SVS is expensive, it is not as expensive as it may initially appear, and there exist rigorous algorithms which may be amenable to reduced-cost, practical adaptations. We will provide more discussion in the final version.
> 3. With or without replacement: this is a good question. Many practical settings involve continuous spaces in which the probability of sampling the same point twice is zero. However, even when the space is discrete we do not expect it to make much difference. It should be slightly better to sample without replacement to avoid redundancy in the mini-batch.
>
> [AGR16]: Anari, Nima, Shayan Oveis Gharan, and Alireza Rezaei. "Monte Carlo Markov chain algorithms for sampling strongly Rayleigh distributions and determinantal point processes." Conference on Learning Theory. PMLR, 2016
>
> [CDV20]: Calandriello, Daniele, Michal Derezinski, and Michal Valko. "Sampling from a k-dpp without looking at all items." Advances in Neural Information Processing Systems 33 (2020): 6889-6899

---

> > ### Author Rebuttal · Reviewer_G6kF · 2026-04-03
> >
> > I did not feel the rebuttal allayed my concerns, but I will continue to judge the paper based on the pre-established merits. (keeping score.)

---

### Official Review · Reviewer_eF9U · 2026-03-09

**Soundness:** 3
**Presentation:** 3
**Significance:** 4
**Originality:** 3
**Overall Recommendation:** 5
**Confidence:** 4

**Summary:**

This paper analyzes the convergence properties of sketch-and-project natural gradient algorithms. Instead of simply analyzing traditional  coupled preconditioned gradient, the authors decouple the gradient and precondition, providing improved rates. The study also reveals that the popular acceleration scheme called SPRING  naturally emerges from accelerated sketch-and-project methods.

**Compliance With Llm Reviewing Policy:**

Affirmed.

**Final Justification:**

The authors' responses demonstrate the potential of their analysis to generalize to non-convex cases, and address my concerns regarding SVS.

**Key Questions For Authors:**

1. The SVS mentioned in the paper is a sampling method that ensures the computed subsampled natural gradient are unbiased. This approach is discrete in nature. Are there continuous sampling methods that also ensure unbiased computation (i.e., using dense sketch matrices rather than sparse 0/1 matrices)? Given the need for extensive computation of determinants, implementing SVS in practice seems computationally intensive.


2. The theoretical analysis requires sampling based on SVS. What would be the impact on the conclusions if SVS is replaced with the more commonly used uniform sampling?

3. The meaning of "acceleration" in Section 6 seems confused. Does it refer to a direct application of well-known acceleration techniques, such as Nesterov acceleration, to solving linear systems?

**Limitations:**

yes

**Strengths And Weaknesses:**

Strengths:

Understanding the superior performance of various forms of natural gradient is an important problem. Compared to existing analyses of preconditioned gradient-type natural gradient methods, this work leverages the  sketch-and-project structure to derive an improved convergence rate.

Weaknesses:

1. The current analysis is restricted to simple quadratic losses and does not explore general or simplified deep learning scenarios. While I acknowledge that analyzing such  non-convex loss landscapes presents fundamental challenges, it would be beneficial to include a discussion on potential pathways for generalizing their proposed techniques to these more complex settings.

2. Regarding the presentation, several mathematical notations are used before they are formally defined, such as the notation $I_S$ below Equation 13  and $\tilde{W}$ around line 214.

---

> ### Author Rebuttal · Authors · 2026-03-30
>
> We thank the reviewer for their thoughtful comments and questions. We first wish to clarify that the novelty of our approach is to treat the *coupled* setting, not the decoupled setting.
>
> **Relevance to deep learning**:
>
> To provide evidence for the relevance of our results to deep learning settings, we have run some additional experiments in which a neural network is trained to solve a small, discrete Poisson problem; see https://figshare.com/s/462ce859778030acf8a2. The results confirm that the relevance of SVS-SNG as a proxy can extend to a simple deep learning setting.
>
> In terms of our theoretical results, we emphasize that our global convergence Theorem 4.2 does apply to general parametric optimization problems including deep learning settings, which already leads to new insights such as that SNG can be convergent when using a single mini-batch per iteration, particularly under squared volume sampling. Moreover, Theorem 5.1 is designed to capture the local convergence properties of SNG for consistent deep learning problems, with consistent LLQ serving as a local quadratic approximation. In fact, the same proof strategy can be extended to yield a local single-step convergence rate for general consistent parametric optimization problems under additional smoothness assumptions. Furthermore, by using a bias-variance decomposition Theorem 5.1 can be extended to case of inconsistent LLQ.  See the response to mmVc for details on these two extensions. We are happy to add a discussion along these lines in the final version.
>
> **Other comments and questions:**
> 1. Mathematical notations: thank you for this feedback. We will clarify these notations in the final version.
> 2. Dense sketching matrices: it is true that directly implementing SVS can be quite expensive. However, the benefit of SVS is that once the sample is selected, the update is efficient. This opens the door to practical schemes which incorporate elements of SVS at a lower cost, such as by developing approximations to rigorous oversample-and-subselect approaches [CDV20] or MCMC approaches [AGR16] to SVS. We believe such directions are more promising than dense sketching matrices, since for dense sketching matrices the update requires accessing the entire Jacobian even once the sketch is generated.
> 3. SVS versus uniform sampling: this is a good question. While in practice uniformly sampled SNG often behaves in line with SVS-SNG, in theory it is possible for SNG to be divergent under uniform sampling, even for the consistent LLQ problems. This was our main motivation for introducing SVS. If it it is helpful we can add an appendix with some toy examples demonstrating this phenomenon in the final version.
> 4. Acceleration: we have perhaps overloaded the term by using it to refer both (1) to the fact that SPRING converges more rapidly than SNG, and (2) to Nesterov-accelerated sketch-and-project methods as initially proposed by [RT20]. The context is that while SPRING has been well-documented to achieve faster convergence than SNG, there has been no previous theoretical explanation for its structure or performance. Our contribution is to show that SPRING can be derived from known Nesterov-accelerated sketch-and-project methods for linear systems. This yields a new interpretation of SPRING and also the first convergence guarantee for SPRING, in the consistent linear least-squares setting. We will clarify these points in the final version.
>
> [YLG22]: Yuan, Rui, Alessandro Lazaric, and Robert M. Gower. "Sketched Newton--Raphson." SIAM Journal on Optimization 32.3 (2022): 1555-1583.
>
> [RT20]: Richtárik, Peter, and Martin Takác. "Stochastic reformulations of linear systems: algorithms and convergence theory." SIAM Journal on Matrix Analysis and Applications 41.2 (2020): 487-524.
>
> [CDV20]: Calandriello, Daniele, Michal Derezinski, and Michal Valko. "Sampling from a k-dpp without looking at all items." Advances in Neural Information Processing Systems 33 (2020): 6889-6899
>
> [AGR16]: Anari, Nima, Shayan Oveis Gharan, and Alireza Rezaei. "Monte Carlo Markov chain algorithms for sampling strongly Rayleigh distributions and determinantal point processes." Conference on Learning Theory. PMLR, 2016

---

> > ### Author Rebuttal · Reviewer_eF9U · 2026-04-01
> >
> > Thanks for the response. My questions have been addressed, and I am happy to raise my score accordingly.

---

### Official Review · Reviewer_mmVc · 2026-03-09

**Soundness:** 2
**Presentation:** 2
**Significance:** 2
**Originality:** 2
**Overall Recommendation:** 4
**Confidence:** 4

**Summary:**

The papers main contribution is to show that the subsampled natural gradient method can be viewed as a variant of the sketch-and-project method applied to a shifted stationarity condition (I’ll define this below). They use this connection to establish convergence of subsampled natural gradient on consistence least squares problem, when using volume sampling. They then show that an accelerated variant of subsampled natural gradient called SPRING, can also be seen as a variant of the accelerated sketch and project method, but this connection does not (yet) yield any rates.

**Compliance With Llm Reviewing Policy:**

Affirmed.

**Final Justification:**

I have decided to increase my score to 4 after the response, because the authors are mostly able to address the issues I raised in terms of clarity, and discussion with other reviewers emphasizing applications of Natural gradient descent for training PINs models, and neural quantum states. But these are not applications I'm an expert in, nor are they part of the core contributions of the paper. I remain uncertain around the technical contribution, which from my view point, is a method for solving least squares (or a linear system), which uses combination of sketch and project, with a fixed point reformulation, in such a way to avoid having to invert part of the linear operator. This is neither trivial, nor significant technical contribution, but given how it fits within the application of the paper, is a reasonable contribution.

**Key Questions For Authors:**

[Excessive unused formalism]. Formalism around $\mathcal{H}$. You introduce $v_{\theta}$ as belonging to Hilbert space, and the loss taking values from this Hilbert space. But all of your theoretical contributions rely on $\mathcal{H} = \mathbb{R}^m.$ Moreover, if you wanted to still talk about Hilbert spaces in the main text, you would need to be more careful.

[Unsupported claims] “These finds in turn yield new insights into small-sample settings,... suggesting that the advantage of SNG over SGD is that it can more effectively exploit spectral decay in the model Jacobian”. This is one of the main claims insights of your re-interpretation of SNG. This claim is not well supported enough in the paper. You mention on lines 307-310 that “$\alpha$ can scale superlinearly in the sample size when $J$ exhibits fast spectral decay (see section 3.3)”. Section 3.3 does not support this claim either. Could you clarify this? This is one of the few insights you get from your perspective, so it needs to be well supported

[Figure 2]. What is the “realistic” and “previous” method? Only your (SVS) method is clearly defined here.

Minor Issues:

Section 3.2: There are some things in this section that have not been carefully defined. For instance, you start by introducing $S_i\in \{1,..,m\}$, then defining $S$ as a set of such $S_i$ indices. You should even defined this square bracket notation where $r[S_i]$ means, get the $S_i$ element of this vector. I suspect what is preventing you from being clear here, is again trying to hold on to this Hilbert space notation $\mathcal{H}$, even though you never use this level of generality.

Equation 3. It’s a bit odd to use pseudo-inverse notation for invertible matrices. But it’s not wrong.

Equation (11). You need to define $\lambda^+\_{\min}(A)$ as the smallest non-zero eigenvalue . This is not standard notation. Also, just a small bibliographic correction, this result in (11) with $\lambda^+\_{\min}(\bar{P})$ first appeared in follow-up work (SDA) and not in the (Gower & Richtarik 2015) paper you cited.

[SDA] Robert Mansel Gower & Peter Richtarik, Stochastic Dual Ascent for Solving Linear Systems, 2015



Line 307: “We emphasize that the suggested mechanism is interpretable under much more general conditions, since $\alpha$ and $\gamma$ can be defined according to any chosen sampling distribution.” This sentence should be removed. Outside of SVS sampling, $\alpha$ and $\gamma$ have no established link to your convergence, so this is bit misleading.

Proposition 5.2 I have no idea what notation $[H, JJ^T] =0$ means.  I had to check your proof in the appendix to find that to find that it implies that $range(HJ) = range(J)$, though this is still not the definition.

Lemma A.2, you should recall where the matrix D is defined, that is, in equation (23). Also this notation matlab notation $D[1:e ,1:e]$ is not totally standard. It’s worth defining.

**Limitations:**

The theory is constrained to a very specialized setting (consistent least squares) and under volume sample (too expensive in practice).

**Strengths And Weaknesses:**

Strength. Identifying SNG (and SPRING) as an instance of accelerated Sketch-and-Project (accelerated), in this special case of a consistent linear least squares, is interesting. Establishing convergence for applying sketch and project to your “shifted stationarity” condition (more on this below) is also interesting.

Weakness. I would say the main weakness is having either very specialized convergence results, in the consistent linear least squares setting which is really solving a linear system. Or having very weak convergence results (Theorem 4.2), more on this below.


Your Theorem 4.2 establishes an asymptotic convergence, and only of the average of gradients. This is really only a sanity check, and gives indication if this method is a good idea. Compare this to the $O(1/\sqrt{T})$ convergence of stochastic gradient descent which holds under weaker assumptions. That is, SGD only requires assuming the gradients are Lipschitz (as you have assumed), but doesn’t require that the function be a composition, or that $v_{\theta}$ be Lipschitz, or have gradients bounded, see Theorem 5.12 in [Handbook] for example. Furthermore, in some sense you are applying a sketching and projecting method for solving the nonlinear system of equations given by $\nabla_{\theta} \mathcal{L}(v_{\theta}) =0.$ There are already convergence results in the nonlinear setting for sketch and project, see [Yuan2022] below. Though, I understand their method is not the same as SNG, since they do not leverage the composite structure. Still, it is the most closely related point in the literature on sketch and project applied to the nonlinear setting.


The remaining of your propositions are for solving a consistent least squares problem, in other words, the linear system $HJ\theta  =b.$  I find it interesting, and a little bit confusing. Because instead of applying sketch and project directly to the stationarity equation $HJ\theta  =b,$ you apply it to $J\theta  = J \theta_t - \eta (H J \theta_t-b).$ Of course, the fixed points of this equation are exactly $\theta$ such that  $HJ \theta -b$. So in a sense, you are re-writing the stationarity as:
$$J\theta  = J \theta - \eta (H J \theta-b)$$
Then fixing $\theta = \theta_t$ on the right hand side, as you would in a fixed point method, then solving for the left hand side using Sketch and project. Because of this initial change, I can see you can no longer immediately import the theory of Sketch and project, and instead need to do some work to prove the convergence in Theorem 5.1. The resulting rate of convergence in Theorem 5.1 is much slower than just standard Sketch and project. As I see it, this is the price you pay for not having direct access to sketches and pseudo inverses of $HJ$, and instead having only access to sketches and pseudo inverses of $J$.




[Handbook] Garrigos, Handbook of Convergence Theorems
for (Stochastic) Gradient Methods, arXiv:2301.11235

[Yuan2022] Rui Yuan et at, Sketched Newton--Raphson, SIAM Journal on Optimization, Vol. 32, Iss. 3, 2022

---

> ### Author Rebuttal · Authors · 2026-03-30
>
> We thank the reviewer for their critical engagement and detailed feedback. While we agree that our results have limitations, we emphasize that to our knowledge, the only previously analyzed SNG algorithm is the two mini-batch variant (clarified below). Since this algorithm is ineffective at small sample sizes in both theory and practice, it has remained an important open problem to explain the growing empirical evidence that SNG can be effective at small sample sizes. By interpreting SNG as a sketch-and-project method, establishing SVS-SNG as a rigorous alternative to the two mini-batch algorithm (Theorem 4.2), and showing that it can lead to meaningful convergence rates in small-sample settings (Theorem 5.1), we make progress on this problem.
>
> We also emphasize that while the consistent LLQ setting of Theorem 5.1 can be viewed as a linear system $HJ\theta = b$, its significance comes from its connection to practically relevant parametric optimization problems. For example note that Theorem 5.1 admits a natural perturbative extension which states that for any consistent parametric optimization problem and any $\epsilon > 0$, for $\theta_t$ close enough to $\theta^*$ and appropriate $\eta$, it holds under relevant smoothness assumptions
>
> $
> \mathbb{E} [ || \theta_{t+1} - \theta^* || ^2_{\tilde{W}^+} | \theta_t ] \leq ( 1 - \kappa^{-1}(H) \cdot \alpha / \gamma + \epsilon) || \theta_t - \theta^* ||^2_{\tilde{W}^+}
> $
>
> where $\tilde{W}$, $H$, $\alpha$, $\gamma$ are defined based on the LLQ approximation about $\theta^* $. Theorem 5.1 can also be extended to inconsistent LLQ using a bias-variance decomposition (see i.e. Lemma C.1 of [GHL25]), leading to the bound
>
> $
> \mathbb{E} || \theta_t - \theta^* || ^2_{\tilde{W}^+} \leq 2( 1 - \rho)^t || \theta_0 - \theta^* || ^2_{\tilde{W}^+} + \frac{2}{\rho} \mathbb{E} _S || J_S^{+(\lambda)} r^*_S  || ^2 _{\tilde{W}^+}
> $
>
> where $\rho = \kappa^{-1}(H) \alpha/\gamma$ and $r^* = r(\theta^* )$. Via tail averaging or a decaying step size, $\theta^* $ can be recovered with a polynomial rate. Importantly, in the inconsistent case SVS-SNG still targets  $ \theta^* = \rm argmin_\theta \mathcal{L}(v_\theta)$ whereas sketch-and-project on $HJ\theta = b$ at best targets $ \tilde{\theta} = \rm argmin_ \theta || HJ \theta - b || ^2 $. This distinction is significant in applications such as neural wavefunctions in which it is crucial to obtain the smallest possible variational energy $\mathcal{L}(v_\theta)$.
>
> Finally, while we are not yet able to provide an analysis without SVS, we have run some additional experiments in which a small neural network is trained to solve a discrete Poisson problem; see https://figshare.com/s/462ce859778030acf8a2. The results confirm that the relevance of SVS-SNG as a proxy can extend to a simple scientific machine learning setting.
>
> **Other discussion points:**
> 1. Regarding [Yuan2022], we were not aware of this reference. We appreciate the pointer and will add a discussion of this work.
> 2.  Regarding the shifted stationarity we find this to be an interesting interpretation with some limitations. The main limitation is that it does not explain why $J\theta$ is a good shift. This is usually explained by the relation to model space gradient descent as discussed in our Section 3.1.
>
> **Key Questions:**
> 1. Excessive unused formalism: we agree that the Hilbert space formalism is unused and will remove it.
> 2. Unsupported claims: we apologize for the lack of explanation and will address this in the final version. For example eq. (4.2) of [DY24] states that, assuming $\lambda=0$ and ${\rm rank(J)} = n$ and squared volume sampling,
>
>     $
>     \alpha \geq \frac{k - \ell}{k - \ell - 1 + \sum_{j = \ell + 1}^n (\sigma_j / \sigma_n)^2}
>     $
>
>     for any $\ell < k$ with $\sigma_j$ being the singular values of $J$. Thus when $\sigma_j$ decay rapidly choosing i.e. $\ell = k/2$ can yield a bound that grows rapidly with $k$.
>
> 3. Figure 2:
>
>     "Realistic" is our eq. (7) with $S$ sampled uniformly.
>
>     "Previous (2 batch)" is a different stochastic  realization of our eq. (5):
>
>     \begin{equation}
>      \theta_{t+1} = \theta_t - \eta (J_S^\top J_S + \lambda I)^{-1} J_{S'}^\top r_{S'},
>     \end{equation}
>
>     where $S,S'$ are sampled independently and uniformly.
>
>     We apologize for the confusion.
>
> **Minor issues:**
>
> We thank you for the detailed feedback and we will address the points you raise in the final version. Regarding $[H, JJ^\top]$ this is the commutator $[A,B] = AB - BA$.
>
> [DY24]: Dereziński, Michał, and Jiaming Yang. "Solving dense linear systems faster than via preconditioning." Proceedings of the 56th Annual ACM Symposium on Theory of Computing. 2024.
>
> [GHL25]: Goldshlager, Gil, Jiang Hu, and Lin Lin. "Worth their weight: Randomized and regularized block Kaczmarz algorithms without preprocessing." arXiv preprint arXiv:2502.00882 (2025).

---

> > ### Author Rebuttal · Reviewer_mmVc · 2026-04-02
> >
> > Dear Authors, thank you for your answers. Though this new claim of yours that your results in Theorem 5.1 can some how be extended to any consistent parametric optimization problem is very unclear, would need to proven, and I don't appreciate this attempt at stretching of your results. Also, now is not the space to try to introduce new results which would require a proof.
> >
> > But as for my main questions. I appreciate your answers, but I still don't understand question 2 around unsupported claims. How exactly does this inequality in [DY24] justify that SNG is better/faster than SGD for small sample settings? What baseline convergence results of SGD are you comparing SNG to here?

---

> > > ### Author Response · Authors · 2026-04-02
> > >
> > > Thanks for your follow-up. We agree that our previous response has somewhat overstepped the scope of the rebuttal by discussing extensions beyond what is proved in the submission. Our intention was only to provide context on how the consistent linear system setting connects to broader optimization settings, but we agree that this is not the right place to introduce additional claims without proof. We therefore retract those extensions from the rebuttal and will not rely on them in the paper.
> > >
> > > Regarding question 2, the intended comparison is not a universal statement that SNG is always faster than SGD. Rather, the comparison concerns how the best provable linear convergence rate scales with the sample size $k$, particularly in strongly convex settings such as the LLQ setting that we consider. In such settings the standard benchmark for mini-batch SGD is that the linear convergence rate scales at best linearly in the sample size; see [J18] for a detailed discussion of the least-squares setting and [Section 6, GG23] for similar results in a sum-of-functions setting, where the linear scaling follows from the fact that the effective smoothness parameter $L_k$ decays no faster than $1/k$ (Lemma 6.5), which in turn allows the step size to increase at most linearly in $k$ (Theorem 6.12).
> > >
> > > In contrast, our Theorem 5.1 shows that for SVS-SNG the linear convergence rate is governed by $\alpha / \gamma$, and [DY24] provides a lower bound on $\alpha$ for which, under sufficiently fast spectral decay of $J$, the predicted dependence of $\alpha$ on $k$ can be stronger than linear. Thus, provided that $\gamma$ behaves benignly, our result identifies a regime in which the predicted $k$-dependence for SVS-SNG is stronger than the usual linear-in-$k$ benchmark for mini-batch SGD. Importantly, the practical regime of interest is when $1 \ll k \ll n$  (for example $k=10^3$, $n=10^6$) rather than the single-sample case $k=1$, so the stronger dependence on $k$ is relevant to realistic small-sample settings.
> > >
> > > [J18]: Jain, Prateek, et al. “Parallelizing stochastic gradient descent for least squares regression: mini-batching, averaging, and model misspecification.” Journal of machine learning research 18.223 (2018): 1-42.
> > >
> > > [GG23]: Garrigos, Guillaume, and Robert M. Gower. “Handbook of convergence theorems for (stochastic) gradient methods.” arXiv preprint arXiv:2301.11235 (2023).

---

### Official Review · Reviewer_zxdt · 2026-03-13

**Soundness:** 3
**Presentation:** 3
**Significance:** 3
**Originality:** 3
**Overall Recommendation:** 5
**Confidence:** 4

**Summary:**

This paper studies subsampled natural gradient descent (SNG) through a sketch-and-project lens
rather than the usual stochastic-preconditioning lens. The main idea is to replace the standard
two-independent-mini-batch proxy with a squared volume sampling (SVS) proxy under which the
expected SNG step remains a preconditioned gradient step despite gradient/preconditioner
coupling. The paper then proves a global convergence guarantee for SVS-SNG
(Theorem 4.2), derives a linear convergence rate for consistent linear least-quadratics (LLQ)
with rate governed by $\alpha/\gamma$ (Theorem 5.1), and shows that the SPRING momentum scheme
arises naturally from accelerated sketch-and-project applied to the natural gradient subproblem
(Theorem 6.1). The empirical section is synthetic and mainly aims to validate the SVS proxy
and the spectral-decay interpretation.

**Compliance With Llm Reviewing Policy:**

Affirmed.

**Final Justification:**

I maintain my Accept recommendation, as the authors successfully addressed all of my questions in their rebuttal.

**Key Questions For Authors:**

1. The paper's main quantitative claim rests on $\alpha/\gamma$, but $\gamma$ is only
characterized at the extremes $k=1$, $k=m$, or under a strong compatibility condition. Figure 3 suggests benign behavior for Gaussian $J$ with quadratic spectral decay. Could the authors provide
one concrete practically motivated regime where $\gamma = O(1)$ can be established?

2. For Conjecture 6.2, can the authors clarify what the main technical obstacle is to proving
the accelerated rate in the LLQ setting?

3. The paper positions SVS as a theoretical proxy and explicitly says it does not recommend direct implementation, yet the conclusions section suggests that SVS could inspire practical sampling strategies. Could the authors clarify what a realistic path to SVS-inspired sampling looks like in the NNW or PINN setting and expand on this point in the manuscript?

**Limitations:**

Yes

**Strengths And Weaknesses:**

### Soundness

The technical derivations appear coherent and sound. The key identity of Lemma 4.1
follows from known expectation formulas for determinantal point processes, Theorem 4.2 reduces to standard stochastic approximation conditions, and Theorem 6.1 is a satisfying algebraic reformulation of SPRING via an explicit variable transformation. I don't see any major issues.

The main limitation is that the central results are proved for SVS-SNG rather than the
practical uniform-mini-batch algorithm. The paper is honest about this and the empirical
proxy-validation figures are supportive, although this gap remains. The $\gamma$ quantity in Theorem
5.1 is also only characterized in special cases, which limits the quantitative content of the
rate result in practice. Both points are acknowledged by the authors.

### Presentation

Clearly written and well organized. The contrast with the prior mini-batch analysis is
communicated effectively and the figures support well with the theoretical claims. The appendix
is sufficiently detailed.

### Originality

The combination of SVS-based analysis of coupled SNG updates and the derivation of SPRING from
accelerated sketch-and-project is novel to my knowledge.

### Significance

The reframing of SNG as sketch-and-project is a meaningful contribution, and the
SPRING reinterpretation is valuable for an Algorithm that has recently been shown useful for Neural network wavefunctions and PINN opimization.

---

> ### Author Rebuttal · Authors · 2026-03-30
>
> We thank the reviewer for their thoughtful comments and questions. Regarding the SVS assumption, we have run some additional experiments in which a small neural network is trained to solve a discrete Poisson problem; see https://figshare.com/s/462ce859778030acf8a2. The results confirm that the relevance of SVS-SNG as a proxy can extend to a simple scientific machine learning setting.
>
> **Key questions:**
> 1. Regarding $\gamma$ we believe that it is challenging to provide strong bounds under realistic conditions, though we hope future works can do so. For now we can identify one additional case in which $\gamma = O(1)$. In particular, if we assume that  $JJ^\top = I$, so that each entry of the model $v_\theta$ is linearly mapped to an orthonormal direction in the parameter space, then it follows that under uniform sampling, $\overline{P} = k /((1 + \lambda) m) J^\top J$, $\tilde{W} = k / ((1+\lambda) m) I$, and
>     $\gamma = 1 / (1+ \lambda)$. Furthermore, in this setting squared volume sampling becomes equivalent to uniform sampling and so the same result holds under SVS. This gives us another sanity check that under benign conditions $\gamma$ can behave benignly.
> 2. Regarding Conjecture 6.2, the main technical obstacle lies in finding an appropriate Lyapunov function which obtains a steady descent when applying SPRING to LLQ. At the moment our Theorem 5.1 shows descent for the solution error in a carefully chosen norm, while existing proofs of accelerated sketch-and-project methods show descent for a somewhat more complicated Lyapunov function. It is not clear how to combine these ideas into a single Lyapunov function that achieves a steady descent in the presence of both $H \neq I$ and acceleration.
> 3. Regarding realistic paths towards SVS-inspired sampling, we see two types of approaches. The first would be to initially sample a larger mini-batch using a standard sampling method, then subselect with squared volume sampling; see for example [CDV20]. The second would be to modify the underlying MCMC chain to incorporate some aspects of squared volume sampling; see for example [AGR16]. The first approach would be suitable for any application, while the second would be more suitable for cases such as NNWs in which there is already MCMC sampling. In either case the theoretical algorithms would likely need to be modified to make them more practical. For example in the first case we might only oversample by a small constant factor rather than by the adaptive factor that is required to guarantee exact squared volume sampling. Hence why in the end we would do "SVS-inspired" sampling rather than true SVS.
>
> We will clarify these points in the final version.
>
> [CDV20]: Calandriello, Daniele, Michal Derezinski, and Michal Valko. "Sampling from a k-DPP without looking at all items." *Advances in Neural Information Processing Systems* 33 (2020): 6889-6899
>
> [AGR16]: Anari, Nima, Shayan Oveis Gharan, and Alireza Rezaei. "Monte Carlo Markov chain algorithms for sampling strongly Rayleigh distributions and determinantal point processes.'" *Conference on Learning Theory*. PMLR, 2016

---

> > ### Author Rebuttal · Reviewer_zxdt · 2026-03-31
> >
> > Thank you for addressing my questions. I have read the rebuttal and decided to keep my positive score.

---

### Decision · Program_Chairs · 2026-04-30

**Decision:**

Accept (regular)

**Comment:**

This paper analyzes "Subsampled natural gradient descent" from the lens of sketch-and-project, providing convergence bounds when using subsampling. Reviews are mostly positive, with only one reviewer recommending rejection due to issues with originality, which he views as low since the paper mainly combines existing ingredients. I have read the author's rebuttal to this criticism, and the paper, and I tend to agree with the authors.